# Mathematical relationships between spinal motoneuron properties

**Arnault H Caillet[1], Andrew TM Phillips[1], Dario Farina[2†], Luca Modenese[1,3]\*†**

[1]Department of Civil and Environmental Engineering, Imperial College London, London, United Kingdom; [2]Department of Bioengineering, Imperial College London, London, United Kingdom; [3]Graduate School of Biomedical Engineering, University of New South Wales, Sydney, Australia

**Abstract** Our understanding of the behaviour of spinal alpha-motoneurons (MNs) in mammals partly relies on our knowledge of the relationships between MN membrane properties, such as MN size, resistance, rheobase, capacitance, time constant, axonal conduction velocity, and afterhyperpolarization duration. We reprocessed the data from 40 experimental studies in adult cat, rat, and mouse MN preparations to empirically derive a set of quantitative mathematical relationships between these MN electrophysiological and anatomical properties. This validated mathematical framework, which supports past findings that the MN membrane properties are all related to each other and clarifies the nature of their associations, is besides consistent with the Henneman's size principle and Rall's cable theory. The derived mathematical relationships provide a convenient tool for neuroscientists and experimenters to complete experimental datasets, explore the relationships between pairs of MN properties never concurrently observed in previous experiments, or investigate inter-mammalian-species variations in MN membrane properties. Using this mathematical framework, modellers can build profiles of inter-consistent MN-specific properties to scale pools of MN models, with consequences on the accuracy and the interpretability of the simulations.

**\*For correspondence:**
l.modenese@unsw.edu.au

†These authors contributed equally to this work and share the senior authorship

**Competing interest:** The authors declare that no competing interests exist.

## Editor's evaluation

This article will be of interest to scientists studying the contribution of motoneuron behaviour to motor control as well as anyone interested in the relation between neuron morphology, intrinsic properties, and neuron behaviour. The authors have distilled decades of research on motoneuron properties into a set of mathematical relationships that can guide both experimentalists and modellers interested in developing realistic models of populations of motoneurons. In fact, Caillet et al. present a data-driven regression analysis to infer the relationships between morphological and electrophysiological measures from spinal motor neurons in different animal species. Finally, the authors emphasize the value of this approach, but also carefully consider its limitations, including inter-study variability and limited sample sizes in the experimental datasets used to derive the relationships between multiple intrinsic properties.

## Introduction

Assessing the morphological and electrophysiological properties of individual spinal alpha-motoneurons (MNs) is crucial for understanding their recruitment and discharge behaviour, and exploring the neuromechanical interplay that controls voluntary motion in mammals. As measured in individual spinal MNs in experimental studies and reported in review papers (*Henneman, 1981*; *Burke, 1981*; *Binder et al., 1996*; *Powers and Binder, 2001*; *Olinder et al., 2006*; *Heckman and Enoka, 2012*), significant correlations exist in mammalian MN pools between the MN morphological and/or electrophysiological properties reported

**eLife digest** Muscles receive their instructions through electrical signals carried by tens or hundreds of cells connected to the command centers of the body. These 'alpha-motoneurons' have various sizes and electrical characteristics which affect how they transmit signals. Previous experiments have shown that these properties are linked; for instance, larger motoneurons transfer electrical signals more quickly. The exact nature of the mathematical relationships between these characteristics, however, remains unclear. This limits our understanding of the behaviour of motoneurons from experimental data.

To identify the equations linking eight motoneuron properties, Caillet et al. analysed published datasets from experimental studies on cat motoneurons. This approach uncovered simple mathematical associations: in fact, only one characteristic needs to be measured experimentally to calculate all the other properties. The relationships identified were also consistent with previously accepted approaches for modelling motoneuron activity. Caillet et al. then validated this mathematical framework with data from studies on rodents, showing that some of the equations hold true for different mammals.

This work offers a quick and easy way for researchers to calculate the characteristics of a motoneuron based on a single observation. This will allow non-measured properties to be added to experimental datasets, and it could help to uncover the diversity of motoneurons at work within a population.

in *Table 1*. For example, the soma size ($D_{soma}$) and the current threshold for spike initiation ($I_{th}$) of an MN are both positively correlated to the axonal conduction velocity ($ACV$) and afterhyperpolarization duration ($AHP$). Also, $I_{th}$ decreases with increasing input resistance ($R$) and $AHP$, $R$ is negatively correlated to $ACV$ and $D_{soma}$, and $ACV$ varies inversely with $AHP$. Besides supporting and sequentially leading to extensions of the Henneman's size principle (*Henneman, 1957*; *Wuerker et al., 1965*; *Henneman et al., 1965a*; *Henneman et al., 1965b*; *Henneman et al., 1974*; *Henneman, 1981*; *Henneman, 1985*), these empirical associations find strong consistency with Rall's theoretical approach of representing the soma and the dendritic trees of MNs as an equivalent cylinder (*Rall, 1977*; *Rall et al., 1992*; *Powers and Binder, 2001*). In such case, MNs behave like resistive-capacitive (RC) circuits, as successfully simulated with computational RC models (*Izhikevich, 2004*; *Dong et al., 2011*; *Negro et al., 2016*; *Teeter et al., 2018*), that rely on further property associations, such as the MN membrane time constant $\tau$ precisely equalling the product of the MN membrane-specific capacitance $C_m$ and resistivity $R_m$ ($\tau = R_m C_m$).

Yet, because the empirical correlations between MN properties were obtained from scattered data from individual experimental studies, the quantitative mathematical associations between the MN properties reported in *Table 1*, beyond their aforementioned global relative variations, remain unclear. For example, the negative correlation between $R$ and $ACV$ was described in the literature with linear (*Fleshman et al., 1981*), exponential (*Burke, 1968*) or power relationships (*Kernell, 1966*; *Kernell and Zwaagstra, 1981*; *Ulfhake and Kellerth, 1984*; *Gustafsson and Pinter, 1984a*), while *Zwaagstra and Kernell, 1980* reported for this association a slope twice greater than *Gustafsson and Pinter, 1984a* in the double-logarithmic space. This makes it difficult to reconcile the conclusions of multiple empirical studies that investigated different property associations. For example, *Burke, 1968* and *Zwaagstra and Kernell, 1980*, respectively, proposed exponential $R - ACV$ and linear $ACV - AHP$ associations, suggesting a hybrid exponential relationship between $R$ and $AHP$, that is different from the power $R - AHP$ relationship directly reported by *Ulfhake and Kellerth, 1984*. These divergences between studies are a major limitation for our understanding of the associations and the distribution of MN properties in an MN pool, which cannot be directly investigated experimentally either, as measuring multiple MN morphometric properties in vitro and electrophysiological properties in vivo for a large sample of MNs is challenging.

Here, we reprocessed and merged the published data from 19 available experimental studies in adult cat preparations to derive and validate a unique set of mathematical power relationships between all pairs of the MN morphometric and electrophysiological cat properties listed in *Table 1*. The significance of these quantitative relationships, which are consistent with the conclusions from Rall's cable theory, supports the notion that the properties reported in *Table 1* are all associated to each other. The uniqueness of these mathematical relationships tackles the aforementioned inter-study variability in describing the data and clarifies our understanding of the quantitative association between spinal MN properties, including for pairs

**Table 1.** The motoneuron (MN) and muscle unit (mU) properties investigated in this study with their notations and SI base units.

$S_{MN}$ is the size of the MN. As reproduced in *Table 2*, the MN size $S_{MN}$ is adequately described by measures of the MN surface area $S_{neuron}$ and the soma diameter $D_{soma}$. $R$ and $R_m$ define the MN-specific electrical resistance properties of the MN and set the value of the MN-specific current threshold $I_{th}$ (*Binder et al., 1996*; *Powers and Binder, 2001*; *Heckman and Enoka, 2012*). $C$ and $C_m$ (constant among the MN pool) define the capacitance properties of the MN and contribute to the definition of the MN membrane time constant $\tau$ (*Gustafsson and Pinter, 1984a*; *Zengel et al., 1985*). $\Delta V_{th}$ is the amplitude of the membrane voltage depolarization threshold relative to resting state required to elicit an action potential. $I_{th}$ is the corresponding electrical current causing a membrane depolarization of $\Delta V_{th}$. AHP duration is defined in most studies as the duration between the action potential onset and the time at which the MN membrane potential meets the resting state after being hyperpolarized. ACV is the axonal conduction velocity of the elicited action potentials on the MN membrane. $S_{mU}$ is the size of the mU. As indicated in *Table 2*, the mU size $S_{mU}$ is adequately described by measures of (1) the sum of the cross-sectional areas (CSAs) of the fibres composing the mU $CSA^{tot}$, (2) the mean fibre CSA $CSA^{mean}$, (3) the innervation ratio IR, that is, the number of innervated fibres constituting the mU, and (4) the mU tetanic force $F^{tet}$. $F^{tw}$ is the MU twitch force.

|  | Properties | Notation | Unit |
| --- | --- | --- | --- |
|  | Size:<br>Neuron surface area<br>Soma diameter | $S_{MN}$<br>$S_{neuron}$<br>$D_{soma}$ | <br>$[\text{m}^2]$<br>$[\text{m}]$ |
|  | Resistance | $R$ | $[\Omega]$ |
|  | Specific resistance per unit area | $R_m$ | $[\Omega \cdot \text{m}^2]$ |
|  | Capacitance | $C$ | $[\text{F}]$ |
|  | Specific capacitance per unit area | $C_m$ | $[\text{F} \cdot \text{m}^{-2}]$ |
|  | Time constant | $\tau$ | $[\text{s}]$ |
|  | Rheobase (current recruitment threshold) | $I_{th}$ | $[\text{A}]$ |
|  | Voltage threshold | $\Delta V_{th}$ | $[\text{V}]$ |
|  | Afterhyperpolarization duration | $AHP$ | $[\text{s}]$ |
| MN properties | Axonal conduction velocity | $ACV$ | $[\text{m} \cdot \text{s}^{-1}]$ |
|  | Size: | $S_{mU}$ |  |
|  | Total fibre cross-sectional area | $CSA^{tot}$ | $[\text{m}^2]$ |
|  | Mean fibre cross-sectional area | $CSA^{mean}$ | $[\text{m}^2]$ |
|  | Innervation ratio | $IR$ | $[]$ |
|  | Tetanic force | $F^{tet}$ | $[\text{N}]$ |
| mU properties | Twitch force | $F^{tw}$ | $[\text{N}]$ |

of properties never simultaneously measured in experiments. These relationships also provide a convenient mathematical framework for modellers for the derivation of appropriate and consistent MN profiles of MN-specific morphometric and electrophysiological properties for the realistic scaling of pools of computational models of MNs, improving the interpretability of model predictions. Using the relationships, experimenters can readily complete their datasets by deriving MN-specific values, which are representative of the literature, for the MN properties that were not experimentally measured.

After deriving the mathematical framework from cat data, we then demonstrate that the normalized cat relationships apply to other mammals by validating them with data from nine adult rat and mouse

electrophysiological studies in vivo. This approach helps better understanding the inter-mammalian-species variations in MN properties. Finally, using additional correlations obtained between some MN and muscle unit (mU) properties from 14 mammal studies, we discuss the empirical relationships obtained between MN properties in the context of the Henneman's size principle.

## Methods
### Selected studies reporting processable adult mammal data
#### Identification of the selected studies
To optimize inter-study consistency, we only selected studies that concurrently measured at least two of the morphometric and/or electrophysiological properties reported in *Table 1* in individual spinal alpha-MNs of healthy adult cats, rats, and mice. To build an extensive set of relevant studies so that the mathematical relationships derived in this study describe a maximum of the data published in the literature, the output of the systematic analysis provided in *Highlander et al., 2020* was screened and used for a further search by reference lists. Among the retrieved studies, larval and postnatal specimens were disregarded because of the pronounced age-related variance (*Highlander et al., 2020*) in the mean values of the electrophysiological properties listed in *Table 1*, which corroborates the non-extrapolability of neonatal neuronal circuitry to older ages (*Song et al., 2006*; *Carp et al., 2008*; *Nakanishi and Whelan, 2010*; *Mitra and Brownstone, 2012*). Due to the nonlinear inter-species scalability of the spinal alpha-MN electrophysiological and morphometric properties (*Manuel et al., 2019*) and a pronounced variance in the inter-species mean values (*Highlander et al., 2020*), non-mammalian species were also disregarded, while the retained data from cats, rats, and mice were processed separately. *Highlander et al., 2020* also report an 'unexplained' variance in the mean electrophysiological property values reported between studies investigating specimens of the same species, sex, and age, which may be explained by the differences between in vivo and in vitro protocols (*Carp et al., 2008*). For this reason, only studies measuring the electrophysiological properties in vivo were considered, while most in vitro studies had already been disregarded at this stage as dominantly performed on neonatal specimens due to experimental constraints (*Manuel et al., 2009*; *Mitra and Brownstone, 2012*). As all but two of the remaining selected studies focussed on lumbosacral MNs, the final set of considered publications was constrained to MNs innervating hindlimb muscles to improve inter-study consistency. From a preliminary screening of the final set of selected studies, it was finally found that relatively more cat studies (19) were obtained than rat (9) and mouse (4) studies, while the cat studies also investigated more pairs of morphometric and electrophysiological properties. The mathematical relationships sought in this article between MN properties were thus derived from the cat data reported in the selected studies, and then validated for extrapolation to rat and mouse data.

#### Selected studies providing cat data: Experimental approaches
The 19 selected studies focussing on cats that were included in this work were published between 1966 and 2001. They applied similar experimental protocols to measure the morphometric and electrophysiological properties reported in *Table 1*. All the selected studies performing morphometric measurements injected in vitro the recorded MNs intracellularly with horseradish peroxidase (HRP) tracer by applying a continuous train of anodal current pulses. The spinal cord was eventually sliced frozen or at room temperature with a microtome or a vibratome and the MN morphometry was investigated. All morphometric measurements were manually performed, and the MN compartments (soma, dendrites, axon) approximated by simple geometrical shapes, yielding some experimental limitations discussed later in 'Methods' and Appendix 1.

To measure the electrophysiological properties, animals were anaesthetized, immobilized, and kept under artificial breathing. Hindlimb muscles of interest were dissected free, and their nerves were mounted onto stimulating electrodes, while maintaining body temperature between 35 and 38°C. After a laminectomy was performed over the lumbosacral region of the spinal cord, the MNs were identified by antidromic invasion following electrical stimulation of the corresponding muscle nerves. All selected cat studies reported the use of electrodes having stable resistances and being able to pass currents up to 10 nA without evident polarization or rectification. Some aspects of the experimental protocol however diverged between the selected studies, such as the age and the sex of the group of adult cats, the size population of cats and recorded MNs, the muscles innervated by the recorded MNs, the means of anaesthesia, the level of oxygen, and whether the spinal dorsal roots were severed (*Kernell, 1966*; *Burke, 1968*; *Barrett and Crill, 1974*; *Fleshman et al., 1981*; *Gustafsson and Pinter, 1984a*; *Gustafsson and Pinter, 1984b*; *Zengel et al., 1985*;

*Foehring et al., 1987*), the complete surrounding hindlimbs were denervated (*Burke, 1968*; *Kernell and Zwaagstra, 1981*), or the spinal cord was transected (*Burke, 1968*; *Krawitz et al., 2001*).

In all the selected cat studies, $ACV$ was calculated as the ratio of the conduction distance and the antidromic spike latency between the stimulation site and the ventral root entry. The studies however did not report how the nerve length was estimated. $AHP$ was measured using brief suprathreshold antidromic stimulations as the time interval between the spike onset to when the voltage deflection of the hyperpolarizing phase returned to the prespike membrane potential. In all studies, $I_{th}$ was obtained as the minimal intensity of a 50–100-ms-long depolarizing rectangular current pulse required to produce regular discharges. $I_{th}$ was obtained with a trial-and-error approach, slowly increasing the intensity of the pulse. The selected studies did not report any relevant source of inaccuracy in measuring $ACV$, $AHP$, and $I_{th}$. The studies measured $R$ using the spike height method (*Frank and Fuortes, 1956*). In brief, small (1–5 nA) depolarizing and/or hyperpolarizing current pulses ($I_{in}$) of 15–100 ms duration were injected through a Wheatstone bridge circuit in the intracellular microelectrode amplifier, and the subsequent change in the amplitude of the membrane voltage potential ($V_m$) was reported in steady-state $V_m - I_{in}$ plots. In all studies, $R$ was obtained by calculating the slope of the linear part of the $V_m - I_{in}$ plots near resting membrane potential by analogy with Ohm's law. It is worth noting for inter-study variability that depolarizing pulses return slightly higher $R$ values than hyperpolarizing currents (*Sasaki, 1991*). To measure the membrane time constant $\tau$, all the selected studies analysed the transient voltage responses ($V_m$) to weak (1–12 nA), brief (0.5 ms), or long (15–100 ms) hyperpolarizing current pulses. Both brief- and long-pulse approaches were reported to provide similar results for the estimation of $\tau$ (*Ulfhake and Kellerth, 1984*). Considering MNs as equivalent isopotential cables, the membrane time constant $\tau$ was identified in all studies as the longest time constant when modelling the measured voltage transient response to a current pulse as the sum of weighted time-dependent exponential terms (*Rall, 1969*; *Rall, 1977*; *Powers and Binder, 2001*). Semi-logarithmic plots of the time history of the voltage transient $V_m$ or of its time derivative $\frac{dV_m}{dt}$ were drawn, and a straight line was fit by eye to the linear tail of the resulting plot (*Fleshman et al., 1988*), the slope of which was $\tau$. A graphical 'peeling' process (*Rall, 1969*) was undertaken to recover the first equalizing time constant $\tau_1$, required to estimate the electronic length ($L$) of Rall's equivalent cylinder representation of the MN and the MN membrane capacitance with Rall's equations (*Rall, 1969*; *Rall, 1977*; *Gustafsson and Pinter, 1984b*; *Powers and Binder, 2001*):

$$L \approx \left[ \frac{\pi}{\left( \frac{\tau}{\tau_1} - 1 \right)} \right]^{\frac{1}{2}}$$

$$C = \frac{\tau}{R} \cdot \frac{L}{\tanh(L)}$$

These electrophysiological measurements were reported to be subject to three main experimental sources of inaccuracy. First, a variable membrane 'leak' conductance, which can be estimated from the ratio of two parameters obtained from the 'peeling' process (*Gustafsson and Pinter, 1984b*), arises from the imperfect seal around the recording micropipette (*Ulfhake and Kellerth, 1984*; *Gustafsson and Pinter, 1984b*; *Pinter and Vanden Noven, 1989*). As reviewed in *Powers and Binder, 2001* and *Olinder et al., 2006*, this 'leak' can affect the measurements of all the properties, notably underestimating the values of $R$ and $\tau$ and overestimating those of $C$. However, this 'leak' is probably not of major significance in the cells of large spike amplitudes (*Gustafsson and Pinter, 1984b*), on which most of the selected studies focussed. Importantly, the membrane 'leak' is also not expected to affect the relations between the parameters (*Gustafsson and Pinter, 1984b*).

Secondly, some nonlinearities (*Ito and Oshima, 1965*; *Burke and ten Bruggencate, 1971*; *Ulfhake and Kellerth, 1984*) in the membrane voltage response to input current steps arise near threshold because of the contribution of voltage-activated membrane conductances to the measured voltage decay (*Fleshman et al., 1988*). This contradicts the initial assumption of the MN membrane remaining passive to input current steps (*Rall, 1969*; *Burke and ten Bruggencate, 1971*). Because of this issue, the ends of the $V_m - I_{in}$ plot are curvilinear and can affect the estimation of $R$, while the transient voltage response to an input current pulse decays faster than exponentially (it undershoots) at the termination of the current injection, which makes it impossible to plot the entire course of $\ln(V(t))$ and challenges the graphical procedure taken to estimate $\tau$ (*Fleshman et al., 1988*). Therefore, some of the selected studies discarded all the MNs that displayed obvious large nonlinearity in the semi-logarithmic $V$–$t$ or $\frac{dV}{dt}$-$t$ plots (*Burke and ten Bruggencate, 1971*). The membrane nonlinearities can be corrected by adding a constant voltage to the entire trace (*Fleshman et al., 1988*) or by considering the three time-constant

model of *Ito and Oshima, 1965* to approximate and subtract to the voltage signal the membrane potential change produced by the current steps. However, none of the selected studies performed such corrections, potentially yielding a systematic underestimation of the values of $R$ and $\tau$ (*Gustafsson and Pinter, 1984b*; *Zengel et al., 1985*; *Powers and Binder, 2001*). Yet, this systematic error for $R$, which should not contribute to inter-study variability, is expected to remain low because the selected studies used input current pulses of low strength (*Ulfhake and Kellerth, 1984*) and measured $R$ from the linear part of the I–V plots near the resting membrane potential where no membrane nonlinearity is expected to occur. This systematic error may also not affect the distribution of recorded $R$ values, as displayed in *Zengel et al., 1985*. *Zengel et al., 1985* besides corrected for the membrane nonlinearities with the three time-constant model approach of *Ito and Oshima, 1965* when measuring $\tau$ and returned values of $\tau$ around 40% higher than the other selected studies. However, this systematic discrepancy is expected to disappear with the normalization of the datasets described in the following, as the reported normalized distributions of $\tau$ are very similar between the selected studies (see density histogram for the $\{\tau; R\}$ dataset in *Appendix 1—figure 1*). Thirdly, the highest source of inaccuracy and inter-study variability arises from the subjective fit 'by eye' of a straight line to the transient voltage when estimating $\tau$ (*Pinter and Vanden Noven, 1989*).

Overall, all the selected studies used similar experimental approaches to measure $ACV$, $AHP$, $I_{th}$, $R$, $C$, and $\tau$, and little sources of inaccuracy and inter-study variability were identified.

## Relationships between MN properties

For convenience, in the following, the notation $\{A; B\}$ refers to the pair of morphometric or electrophysiological MN properties $A$ and $B$, with $A, B \in \{S_{MN}; ACV; AHP; R; I_{th}; C; \tau\}$, defined in *Table 1*. The selected studies generally provided clouds of data points for pairs $\{A; B\}$ of concurrently measured MN properties through scatter graphs. These plots were manually digitized using the online tool WebPlotDigitizer (*Ankit, 2020*). When a study did not provide such processable data – most reported the mean ± sd property values of the cohort of measured MNs – the corresponding author was contacted to obtain the raw data of the measured MN properties, following approval of data sharing. Upon reception of data, datasets of all possible $\{A; B\}$ pairs were created and included into the study.

## Normalized space and choice of regression type

The sets of data retrieved from different cat studies for each property pair $\{A; B\}$ were merged into a 'global' dataset dedicated to that property pair. The $\{A; B\}$ data was also normalized for each study and transformed as a percentage of the maximum property value measured in the same study and normalized 'global' datasets were similarly created. A least-squares linear regression analysis was performed for the $\ln(A) - \ln(B)$ transformation of each global dataset yielding relationships of the type $\ln(A) = a \cdot \ln(B) + k$, which were then converted into power relationships of the type $A = k \cdot B^a$ (eq. 1). The adequacy of these global power trendlines and the statistical significance of the correlations were assessed with the coefficient of determination $r^2$ (squared value of the Pearson's correlation coefficient) and a threshold of 0.01 on the p-value of the regression analysis, respectively. For each $\{A; B\}$ pair, the normalized global datasets systematically returned both a higher $r^2$ and a lower p-value than the datasets of absolute $\{A; B\}$ values, in agreement with the 'unexplained' inter-experimental variance reported in *Highlander et al., 2020*. It was therefore decided to investigate the $\{A; B\}$ pairs of MN properties in the normalized space using the normalized global datasets to improve the cross-study analysis. For this preliminary analysis and the rest of the study, power regressions (eq. 1) were preferred to linear ($A = k + a \cdot B$) or exponential ($A = k \cdot e^{a \cdot B}$) regressions to maintain consistency with the mathematical structure of the equations from Rall's cable theory (*Rall, 1957*; *Rall, 1959*; *Rall, 1960*; *Powers and Binder, 2001*) and of other well-defined relationships, such as $R = k \cdot I_{th}^{-1}$. Also, a least-squares linear regression analysis was preliminary performed for each normalized experimental dataset and for its $\ln(A) - \ln(B)$ and $\ln(A) - B$ transformations, yielding linear, power, and exponential fits to the data. The power regressions overall returned $r^2 > 0.5$ for more experimental datasets than the linear and exponential fits (see *Appendix 1—table 1*). To avoid bias, power regressions were the only type of fit used in this study, despite a few datasets being more accurately fitted by linear or exponential regressions (the difference in the quality of the fit was very small in all cases). Other regression types, such as polynomial fits, were not justified by previous findings in the literature.

## Global datasets and data variability

The data variability between the studies constituting the same global dataset $\{A; B\}$ was assessed by analysing the normalized distributions of the properties $A$ and $B$ with four metrics, which were the range of the measured data, the mean of the distribution, the coefficient of variation (CoV) calculated as the standard deviation (sd) divided by the mean of the distribution, and the ratio $\frac{Me}{Md}$ of the median and the mean of the distribution. The inter-study data variability in the $\{A; B\}$ global dataset was assessed by computing the $mean_g \pm sd_g$ across studies of each of the four metrics. Low variability between the data distributions from different studies was concluded if the normalized distributions (in percentage maximum value) returned similar values for the range, mean, CoV, and $\frac{Me}{Md}$ metrics, that is, when $sd_g < 10\%$ and $\frac{sd_g}{mean_g} < 0.15$ were obtained for all four metrics. In such case, the data distributions would respectively span over similar bandwidth length of the MU pool, be centred around similar mean values, and display similar data dispersion and similar skewness. The relative size of the experimental datasets reported by the selected studies constituting the same global datasets was also compared to assess their relative impact on the regression curves fitted to the global datasets. Then, the inter-study variability in associating the distributions of properties $A$ and $B$ was assessed by computing the 95% confidence interval of the linear model fitted to the global dataset $\{A; B\}$ in the log–log space, which yields the power regression $A = k \cdot B^a$. Low inter-study variability was considered if the value of $a$ varied less than 0.4 in the confidence interval.

The variability of the data distribution of a property $A$ between multiple global datasets $\{A; B\}$ was then assessed by computing the same four metrics as previously (range, mean, CoV, $\frac{Me}{Md}$) for each global dataset, and then computing the $mean_G \pm sd_G$ of each of the four metrics across the global datasets $\{A; B\}$. Low variability between the global datasets in the distribution of property $A$ was considered if $sd_G < 10\%$ and $\frac{sd_G}{mean_G} < 0.15$ were obtained for all four metrics. If the inter-study and inter-global dataset variability was low, the global datasets were created and processed to derive the mathematical relationships between MN properties according to the procedure described below.

## Size-dependent normalized relationships

From inspection of the considered cat studies, most of the investigated MN property pairs comprised either a direct measurement of MN size, noted as $S_{MN}$ in this study, or another variable well-admitted to be strongly associated to size, such as $ACV$, $AHP$, or $R$. Accordingly, and consistently with Henneman's size principle, we identified $S_{MN}$ as the reference MN property with respect to which relationships with the electrophysiological MN properties in **Table 1** were investigated. To integrate the data from all global normalized datasets into the final relationships, the MN properties in **Table 1** were processed in a step-by-step manner in the order $ACV$, $AHP$, $R$, $I_{th}$, $C$, $\tau$, as reproduced in **Figure 1** for two arbitrary properties, to seek a 'final' power relationship of the type **eq.1** between each of them and $S_{MN}$. For each electrophysiological property $A$ and each $\{A; B\}$ dataset, we considered two cases. If $B = S_{MN}$, the global dataset was not further processed as the electrophysiological property $A$ was already related to MN size $S_{MN}$ from experimental measurements. If $B \neq S_{MN}$, the global $\{A; B\}$ dataset was transformed into a new $\{A; S_{MN}\}$ dataset by converting the discrete values of $B$ with the trendline regression $S_{MN} = k_d \cdot B^d$, which was obtained at a previous step of the data processing. With this dual approach, as many MN size-dependent $\{A; S_{MN}\}$ intermediary datasets as available $\{A; B\}$ global datasets were obtained for each property $A$. These size-dependent datasets were merged into a 'final' $\{A; S_{MN}\}$ dataset to which a least-squares linear regression analysis was performed for the $\ln(A) - \ln(S_{MN})$ transformation, yielding relationships of the type $\ln(A) = c \cdot \ln(S_{MN}) + k_c$, which were converted into the power relationships:

$$A = k_c \cdot S_{MN}^c$$

The adequacy of these final power trendlines and the statistical significance of the correlations were assessed identically to the $\{A; B\}$ relationships derived directly from normalized experimental data, using the coefficient of determination $r^2$ (squared value of the Pearson's correlation coefficient) and a threshold of 0.01 on the p-value of the regression analysis, respectively. If $r^2 > 0.3$ (i.e., $r > 0.55$) and p<0.01, another power trendline $S_{MN} = k_d \cdot A^d$ was fitted to the final dataset to describe the inverse relationship for $\{S_{MN}; A\}$ and used in the processing of the next-in-line property.

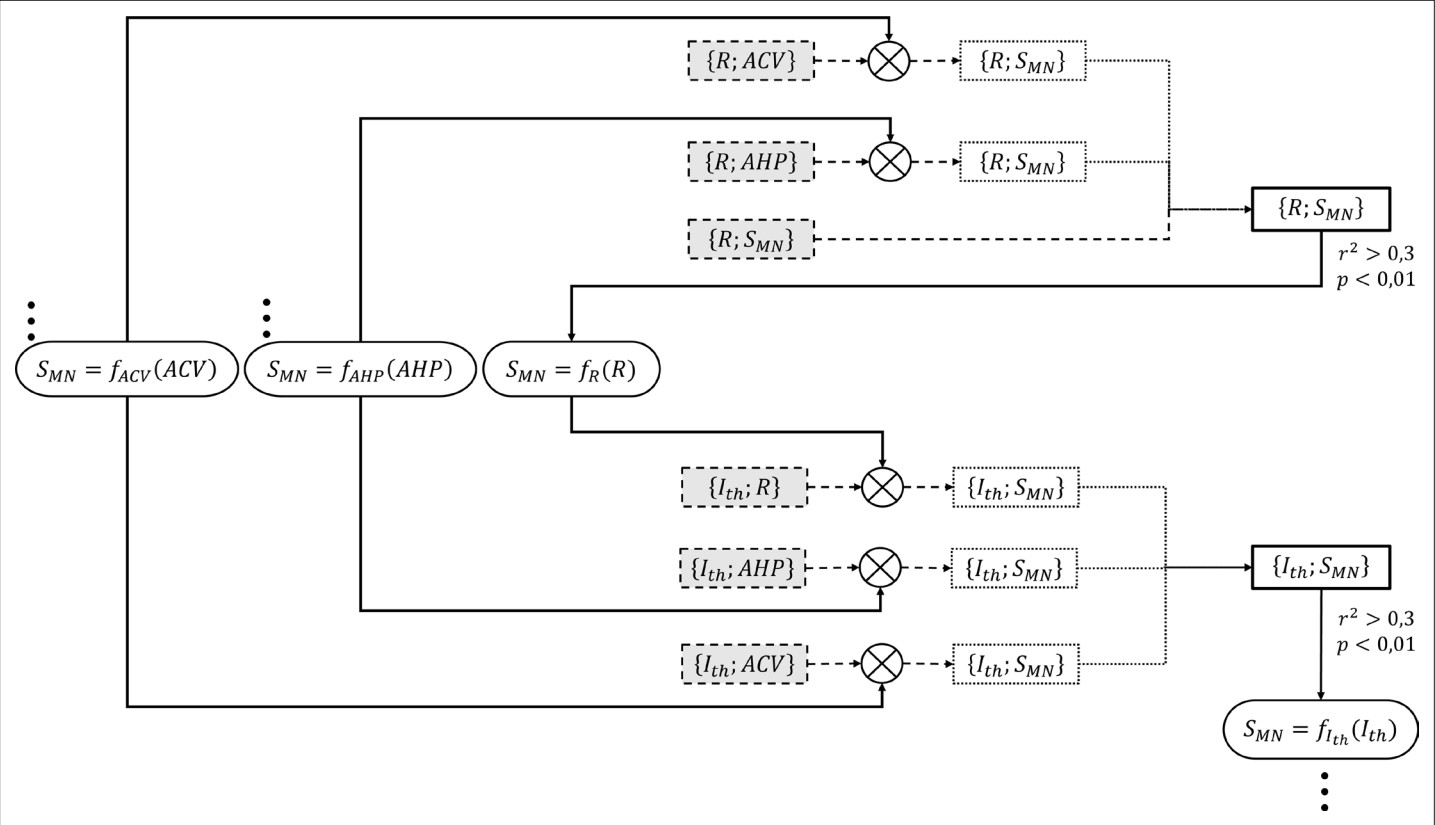

**Figure 1.** Detailed example for the process adopted to successively create the two final $\{R; S_{MN}\}$ and $\{I_{th}; S_{MN}\}$ datasets (right-side thick-solid contour rectangular boxes). These final datasets were obtained from respectively three and three normalized global datasets of experimental data obtained from the literature (dashed-contour grey-filled boxes) $\{\{R; ACV\}, \{R; AHP\}, \{R; S_{MN}\}\}$ and $\{\{I_{th}; R\}, \{I_{th}; AHP\}, \{I_{th}; ACV\}\}$. The $\{R; ACV\}$ and $\{R; AHP\}$ datasets were first transformed ($\otimes$ symbol) into two intermediary $\{R; S_{MN}\}$ datasets (dotted-contour boxes) by converting the $ACV$ and $AHP$ values to equivalent $S_{MN}$ values with two 'inverse' $S_{MN} = f_{ACV}(ACV)$ and $S_{MN} = f_{AHP}(AHP)$ power relationships (oval boxes with triple dots), which had been previously obtained from two unshown steps that had yielded the final $\{ACV; S_{MN}\}$ and $\{AHP; S_{MN}\}$ datasets. The two intermediary $\{R; S_{MN}\}$ datasets were merged with the remaining global $\{R; S_{MN}\}$ dataset to yield the final $\{R; S_{MN}\}$ dataset, to which a power relationship of the form $R = k \cdot S_{MN}^c$ was fitted. If $r^2 > 0.3$ and p<0.01, an 'inverse' power relationship $S_{MN} = f_R(R)$ (oval box) was further fitted to this final dataset. In a similar approach, the three normalized global datasets $\{I_{th}; R\}, \{I_{th}; AHP\}$, and $\{I_{th}; ACV\}$ were transformed with the three 'inverse' relationships into intermediary $\{I_{th}; S_{MN}\}$ datasets, which were merged to yield the final $\{I_{th}; S_{MN}\}$ dataset. An 'inverse' $S_{MN} = f_{I_{th}}(I_{th})$ power relationship was further derived to be used in the creation of the final $\{C; S_{MN}\}$ and $\{\tau; S_{MN}\}$ datasets in the next taken steps.

### Normalized mathematical relationships between electrophysiological properties

When a final relationship with $S_{MN}$ was obtained for two MN properties $A$ and $B$ by the procedure described above, that is, $A = k_{cA} \cdot S_{MN}^{c_A}$ and $B = k_{cB} \cdot S_{MN}^{c_B}$, one of the expressions was mathematically inverted and a third empirical relationship was derived for the property pair $\{A; B\}$ :

$$A = \frac{k_{cA}}{(k_{cB})^{\frac{c_A}{c_B}}} \cdot (B)^{\frac{c_A}{c_B}} = k_e \cdot B^e$$

This procedure was applied to all possible $\{A; B\}$ pairs in **Table 1**.

### Validation of the normalized relationships

The normalized relationships were validated using a standard fivefold cross-validation procedure (**Chollet, 2021**). The data in each $\{A; B\}$ global normalized dataset was initially randomized and therefore made independent from the studies constituting it. Each shuffled global dataset was then split into five non-overlapping complementary partitions, each containing 20% of the data. Four partitions including 80% of the data were taken to constitute a training set from which the normalized

relationships were built as described previously. The latter equations were validated against the last data partition, which includes the remaining 20% of the data and which is called test set in the following. To perform this validation, the final size-related relationships $A = k_c \cdot S_{MN}^c$ and relationships between electrophysiological properties $A = k_e \cdot B^e$ obtained with the training set were applied to the $S_{MN}$ or $B$ data in each test set, yielding predicted values $A$. It was then assessed to what extent the mathematical relationships predicted the test data $A$ by calculating the normalized maximum error ($nME$), the normalized root mean squared error ($nRMSE$) and the coefficient of determination $r_{pred}^2$ between predicted and control values of $A$ in each test set. This process was repeated for a total of five times by permutating the five data partitions and creating five different pairs of one training set and one test set. The $nME$, $nRMSE$, and $r_{pred}^2$ values were finally averaged across the five permutations. For each global dataset, the average $r_{pred}^2$ was compared to the $r_{exp}^2$ obtained from the power trendlines least-squared fitted to the control data in the log–log space. Once validated with this fivefold cross-validation method, the final normalized size-dependent relationships $A = k_c \cdot S_{MN}^c$ and relationships between electrophysiological properties $A = k_e \cdot B^e$ were computed for the complete global datasets.

## Scaling of the normalized relationships

The $A = k_c \cdot S_{MN}^c$ and $A = k_e \cdot B^e$ normalized relationships were finally scaled to typical cat property values in three steps. First, it was assessed from the literature the fold range over which each MN property was to vary. $S_{MN}$ varied over a $q_S^E$ -fold range, taken as the average across cat studies of the ratios of minimum and maximum values measured for $S_{MN}$ :

$$q_S^E = \frac{1}{N_{studies}} \sum_{i=1}^{N_{studies}} \left[ \frac{(S_{MNi})_{max}}{(S_{MNi})_{min}} \right]$$

An experimental ratio $q_A^E$ was similarly obtained for each electrophysiological property $A$. As $A = k_c \cdot S_{MN}^c$, any electrophysiological property $A$ could theoretically vary over a $q_A^T = \left( q_S^E \right)^{|c|}$ -fold range, which was compared to $q_A^E$ for consistency in the results. Then, a theoretical range $\left[ A_{min}; A_{max} \right]$ of values were derived for each electrophysiological property $A$. Defining $A_{min}^E$ and $A_{max}^E$ as the respective average of the minimum and maximum $A$-values across studies, it was enforced $\frac{A_{max}}{A_{min}} = q_A^T$ and $\frac{A_{min} + A_{max}}{2} = \frac{A_{min}^E + A_{max}^E}{2}$ so that the $\left[ A_{min}; A_{max} \right]$ theoretical ranges were consistent with the normalized size-dependent relationships while best reproducing the experimental data from cat literature. A theoretical range for $S_{MN} \left[ (S_{MN})_{min}; (S_{MN})_{max} \right]$ was similarly built over the $q_S^E$ -fold range previously derived. Finally, the intercept $k_c$ in the size-dependent relationships $A = k_c \cdot S_{MN}^c$ was scaled as

$$\begin{cases} k_c = \frac{A_{min}}{(S_{MN})_{min}^c} & ; \quad if \; c > 0 \\ k_c = \frac{A_{max}}{(S_{MN})_{min}^c} & ; \quad if \; c < 0 \end{cases}$$

A similar approach was used to mathematically derive the intercept $k_e$ and scale the relationships $A = k_e \cdot B^e$ between electrophysiological properties.

## Extrapolation to rats and mice

It was finally assessed whether the $A = k_c \cdot S_{MN}^c$ and $A = k_e \cdot B^e$ scaled relationships derived from cat data accurately predicted rat and mouse quantities reported in other experimental studies. These mathematical relationships were applied to the $S_{MN}$ or $B$ data in each $\{A; B\}$ global dataset of absolute values obtained for rats and mice independently. The accuracy in predicting the quantity $A$ was assessed for each available dataset with the same three validation metrics used to validate the normalized power trend lines: the normalized maximum error ($nME$), the normalized root-mean-square error ($nRMSE$) and the coefficient of determination $r^2$ between predicted and experimental values of A.

## Definitions of MN size $S_{MN}$ and mU size $S_{mU}$

### MN size $S_{MN}$

The selected studies that performed both electrophysiological and morphometric measurements on the same MNs dominantly measured the MN soma diameter $D_{soma}$ as an index of MN size. Therefore,

we chose $S_{MN} = D_{soma}$ in the described methodology. In the literature on spinal MNs in adult mammals, the size of an MN $S_{MN}$ is also related to the measures of the somal cross-sectional area $CSA_{soma}$ (**Gao et al., 2009**; **Friese et al., 2009**; **Deardorff et al., 2013**; **Mierzejewska-Krzyżowska et al., 2014**; **Dukkipati et al., 2018**), soma surface area $S_{soma}$ (**Ulfhake and Kellerth, 1984**; **Brandenburg et al., 2020**), axonal diameter $D_{axon}$ (**Cullheim, 1978**), individual dendrite diameter $D_{dendrite}$ (**Amendola and Durand, 2008**; **Carrascal et al., 2009**; **Mantilla et al., 2018**), individual dendritic surface area $S_{dendrite}$ (**Li et al., 2005**; **Obregón et al., 2009**; **Carrascal et al., 2009**; **Filipchuk and Durand, 2012**; **Kanjhan et al., 2015**), total dendritic surface area (**Ulfhake and Cullheim, 1988**; **Amendola and Durand, 2008**; **Brandenburg et al., 2020**), or total MN surface area $S_{neuron}$ (**Burke et al., 1982**) defined as the summed soma and dendritic surface areas.

In the references (**Zwaagstra and Kernell, 1981**; **Ulfhake and Kellerth, 1981**), a linear correlation between $D_{soma}$ and the average diameter of the stem dendrites $D_{dendrite_{mean}}$ has been reported ($r >$ 0.62, population size in {40; 82} cells). Moreover, a linear or quasi-linear correlation has been found (**Cullheim et al., 1987**; **Ulfhake and Cullheim, 1988**; **Moschovakis et al., 1991**; **Prakash et al., 2000**; **Li et al., 2005**; **Obregón et al., 2009**; **Mantilla et al., 2018**) between the stem dendrite diameter $D_{dendrite}$ and the membrane surface area of the corresponding dendritic tree $S_{dendrite}$ ($r >$ 0.78, population size in $[33; 342]$ dendrites). Therefore, from these studies, we can assume an approximate linear correlation between $D_{soma}$ and the average dendritic surface area $S_{dendrite_{mean}}$. Moreover, the number of dendritic trees $N_{dendrites}$ per cell increases with increasing soma surface $S_{soma}$ (**Brandenburg et al., 2018**), and a linear correlation between $D_{soma}$ or $S_{dendrites}$ with $N_{dendrites}$ has been observed (**Zwaagstra and Kernell, 1981**; **Ulfhake and Cullheim, 1988**; **Amendola and Durand, 2008**) ($r >$ 0.40, population size in {14; 32; 87} cells). It was therefore assumed that $D_{soma}$ and total dendritic surface area $S_{dendrite}^{tot}$ are linearly correlated. This assumption/approximation is consistent other conclusions of a linear correlations between $D_{soma}$ and $S_{dendrite}^{tot}$ (**Amendola and Durand, 2008**; **Brandenburg et al., 2020**). Then, according to typical values of $D_{soma}$, $S_{soma}$, and $S_{dendrites}^{tot}$ obtained from the studies previously cited, yielding $S_{dendrites}^{tot} \gg S_{soma}$, it was also assumed $S_{dendrites}^{tot} \approx S_{neuron}$. It is thus concluded that $D_{soma}$ is linearly related to total neuron membrane area $S_{neuron}$, consistently with results by **Burke et al., 1982** ($r = 0.61$, 57 cells).

For the above reasons and assumptions, the mathematical relationships $A = k_c \cdot S_{MN}^c$ derived previously, with $S_{MN} = D_{soma}$, were extrapolated with a gain to relationships between $A$ and $S_{MN} = S_{neuron}$, notably permitting the definition of surface specific resistance $R_m$ and capacitance $C_m$. Following the same method as described above, theoretical ranges $\left[ (S_{neuron})_{min} ; (S_{neuron})_{max} \right]$ were derived from additional morphometric studies on adult cat spinal alpha-MNs. The new relationships were extrapolated as $A = k_c \cdot D_{soma}^c = \frac{(S_{neuron})_{min}}{(D_{neuron})_{min}} \cdot S_{neuron}^c$.

It must however be highlighted that the conclusion of a linear correlation between $D_{soma}$ and $S_{neuron}$, while plausible, is crude. Morphometric measurements of individual MNs are indeed difficult and suffer many limitations. MN staining, slice preparations, and MN reconstruction and identification are complex experimental procedures requiring a large amount of work; the results from most of the cited studies thus rely on relatively small pools of investigated MNs. Most cited studies did not account for tissue shrinkage after dehydration, while some may have failed to assess the full dendritic trees from MN staining techniques (**Brandenburg et al., 2020**), thus underestimating dendritic membrane measurements. Moreover, before automated tools for image segmentation, landmark mapping, and surface tracking (**Amendola and Durand, 2008**; **Obregón et al., 2009**; **Mierzejewska-Krzyżowska et al., 2014**) were available, morphometric measurements were performed from the manual reproduction of the cell outline under microscope, yielding important operator errors reported to be of the order of ~0.5 µm, that is, $\sim$ 20% stem diameter. In these studies, morphological quantities were besides obtained from geometrical approximations of the 2D MN shape, such as modelling the individual dendritic branches as one (**Cullheim et al., 1987**) or more (**Prakash et al., 2000**) equivalent cylinders for the derivation of $S_{dendrite}$. Morphometric measurement approaches also varied between studies; for example, oval or circle shapes were best fitted by sight onto the soma outline and equivalent $D_{soma}$ quantities were derived either as the mean of measured maximum and minimum oval diameters or as the equivalent diameter of the fitted circle. In these studies, $CSA_{soma}$ and $S_{soma}$ were therefore derived by classical equations of circle and spheric surface areas, which contradicts the results by **Mierzejewska-Krzyżowska et al., 2014** obtained from surface segmentation of a linear relationship between $D_{soma}$ and $CSA_{soma}$ ($r = 0.94$, 527 MNs). Finally, no correlation between soma size

**Table 2.** Measurable indices of motoneuron (MN) and muscle unit (mU) sizes in mammals. $S_{MN}$ and $S_{mU}$ are conceptual parameters which are adequately described by the measurable and linearly inter-related quantities reported in this table.

| MN size ($S_{MN}$) | mU size ($S_{mU}$) |
| --- | --- |
| $S_{neuron}$ | $F^{tet}$ |
| $D_{soma}$ | $IR$ |
| | $CSA^{mean}$ |
| | $CSA^{tot}$ |

and $N_{dendrites}$ was found (**Ulfhake and Kellerth, 1981**; **Cullheim et al., 1987**). Therefore, the linear correlation between $D_{soma}$ and $S_{neuron}$ is plausible but crude and requires awareness of several important experimental limitations and inaccuracies. Conclusions and predictions involving $S_{neuron}$ should be treated with care in this study as these morphometric measurements lack the precision of the measures performed for the electrophysiological properties listed in **Table 1**.

## Muscle unit size $S_{mU}$

To enable future comparisons between MN and mU properties, we here assess potential indices of the mU size $S_{mU}$ suggested in the literature. The size of an mU ($S_{mU}$) can be defined as the sum $CSA^{tot}$ of the CSAs of the innervated fibres composing the mU. $CSA^{tot}$ depends on the mU innervation ratio ($IR$) and on the mean CSA ($CSA^{mean}$) of the innervated fibres: $S_{mU} = CSA^{tot} = IR \cdot CSA^{mean}$. $CSA^{tot}$ was measured in a few studies on cat and rat muscles, either indirectly by histochemical fibre profiling (**Burke and Tsairis, 1973**; **Dum and Kennedy, 1980**; **Burke, 1981**), or directly by glycogen depletion, periodic acid Schiff (PAS) staining and fibre counting (**Burke et al., 1982**; **Rafuse et al., 1997**). The mU tetanic force $F^{tet}$ is however more commonly measured in animals. As the fibre mean specific force $\sigma$ is considered constant among the mUs of one muscle in animals (**Lucas et al., 1987**; **Enoka, 1995**), the popular equation $F^{tet} = \sigma \cdot IR \cdot CSA^{mean}$ (**Burke, 1981**; **Enoka, 1995**) returns a linear correlation between $F^{tet}$ and $IR \cdot CSA^{mean} = S_{mU}$ in mammals. Experimental results (**Burke and Tsairis, 1973**; **Bodine et al., 1987**; **Chamberlain and Lewis, 1989**; **Kanda and Hashizume, 1992**; **Rafuse et al., 1997**; **Hegedus et al., 2007**) further show a linear correlation between $F^{tet}$ and $IR$ and between $F^{tet}$ and $CSA^{mean}$. Consequently, $F^{tet}$, $IR$, and $CSA^{mean}$ are measurable, consistent, and linearly related measures of $S_{mU}$ in animals, as summarized in **Table 2**.

## Relationships between MN and mU properties

To assess whether the size-dependent relationships $A = k_c \cdot S_{MN}^c$ derived in this study were in accordance with Henneman's size principle of motor unit recruitment, we identified a set of experimental studies that concurrently measured an MN property $B_{MN}$ (**Table 1**) and an mU property $A_{mU}$ for the same MU. The normalized global datasets obtained for the pairs $\{A_{mU}; B_{MN}\}$ were fitted with power trendlines, as previously described for MN properties, yielding $A_{mU} = k \cdot B_{MN}^b$ relationships. Using both the definition of $S_{mU}$ (**Table 2**) and the $A = k_c \cdot S_{MN}^c$ relationships derived previously, the $A_{mU} = k \cdot B_{MN}^b$ relationships were then mathematically transformed into $S_{mU} = k \cdot S_{MN}^c$ relationships. If all $c$-values obtained from different $\{A_{mU}; B_{MN}\}$ pairs were of the same sign, it was concluded that mU and MN sizes were monotonically related. Considering the limited data available obtained from the literature, data obtained for MNs innervating different hindlimb muscles were merged. Also, cat, rat, and mouse studies were processed independently but the resulting $c$-values were compared without regards to the related species.

## Results

We respectively identified 19, 6, and 4 studies respecting our desired criteria on cats, rats, and mice that reported processable experimental data for the morphometric and electrophysiological properties listed in **Table 1**. Additional publications including some from the past 10 years were identified but could not be included in this work as no processable data could be recovered. An additional 14 studies were found to perform concurrent MN and mU measurements on individual motor units. From the selected cat studies, the 17 pairs of MN properties and the 8 pairs of one MN and one mU property represented in the bubble diagram of **Figure 2A** were investigated.

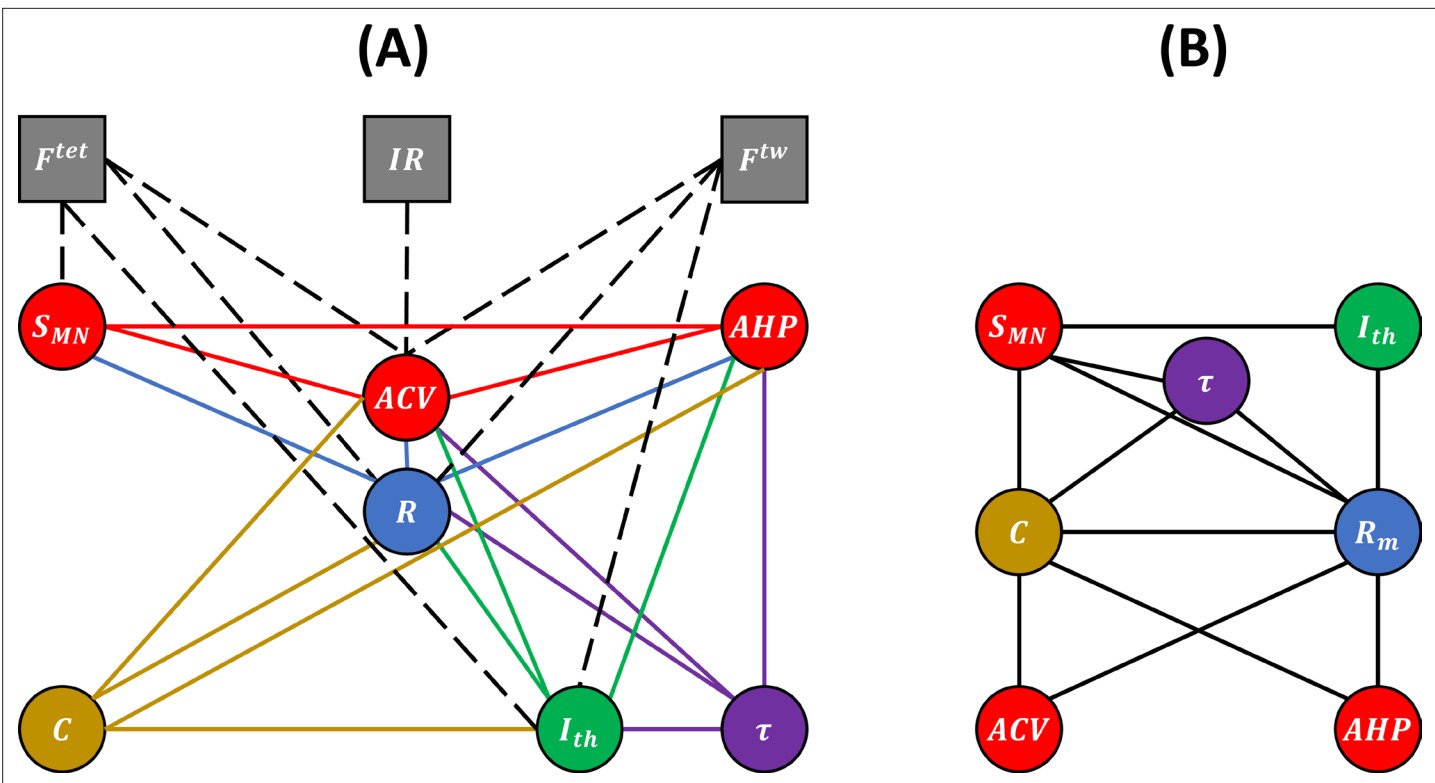

**Figure 2.** Experimental (**A**) and unknown (**B**) relations between motoneuron (MN) and muscle unit (mU) properties. (**A**) Bubble diagram representing the pairs of MN and/or mU properties that could be investigated in this study from the results provided by the 40 studies identified in our web search. MN and mU properties are represented by circle and square bubbles, respectively. Relationships between MN properties are represented by coloured connecting lines; the colours red, blue, green, yellow, and purple are consistent with the order $ACV$, $AHP$, $R$, $I_{th}$, $C$, $\tau$ in which the pairs were investigated (see *Table 3* for mathematical relationships). Relationships between one MN and one mU property are represented by black dashed lines. (**B**) Bubble diagram representing the mathematical relationships proposed in this study between pairs of MN properties for which no concurrent experimental data has been measured to date.

## Relationships between MN properties

### Global datasets and data variability

The experimental data retrieved from the selected studies for the 17 pairs of MN properties drawn in *Figure 2A* were merged into 17 normalized global datasets plotted in *Figure 3*. Eight global datasets $\left(\{ACV; S_{MN}\}, \{AHP; ACV\}, \{R; S_{MN}\}, \{R; ACV\}, \{R; AHP\}, \{I_{th}; R\}, \{I_{th}; ACV\}, \{\tau; R\}\right)$ included the data from two or more experimental studies.

These studies reported in each global dataset similar normalized property distributions, as visually displayed in the frequency histograms in *Appendix 1—figure 1*. This was confirmed by the $mean_g \pm sd_g$ calculations described in 'Methods' of the four metrics range, mean, CoV, and $\frac{Me}{Md}$ displayed as error bars in *Appendix 1—figure 2*. Out of the 64 $mean_g$ +/- $sd_g$ calculations (4 metrics for 16 property distributions of the 8 global datasets), 55 (86%) returned $sd_g < 10\%$ and $\frac{sd_g}{mean_g} < 0.15$. Otherwise, the distributions of only three, two, and four properties showed sensibly higher inter-study variability in the global datasets for the range, mean, and CoV metrics, respectively, however, still verifying $sd_g < 15\%$ and $\frac{sd_g}{mean_g} < 0.33$. Additional details are provided in Appendix 1. Then, as displayed in *Figure 3* and reported in *Table 3*, power trendlines and their 95% confidence interval were fitted to the global datasets. All 17 trendlines were statistically significant and adequately represented by a power model $A = k_a \cdot B^a$ (p-value$<10^{-5}$ and $r^2 \in [0.34; 0.72]$ for 15 datasets; p-value$<10^{-5}$ and $r^2 \in \{0.24; 0.17\}$ for the $\{C; AHP\}$ and $\{C; ACV\}$ datasets, respectively). As displayed in green dotted lines in *Figure 3*, the confidence interval remained narrow for all datasets with the value of power $a$ varying less than 0.4, except for the $\{I_{th}; ACV\}$ dataset, suggesting a low inter-study variability in associating the distributions of two properties. As displayed in *Appendix 1—figure 3*, the selected studies however reported datasets of different sizes. Yet, out of the 35 experimental datasets constituting the 8 global datasets

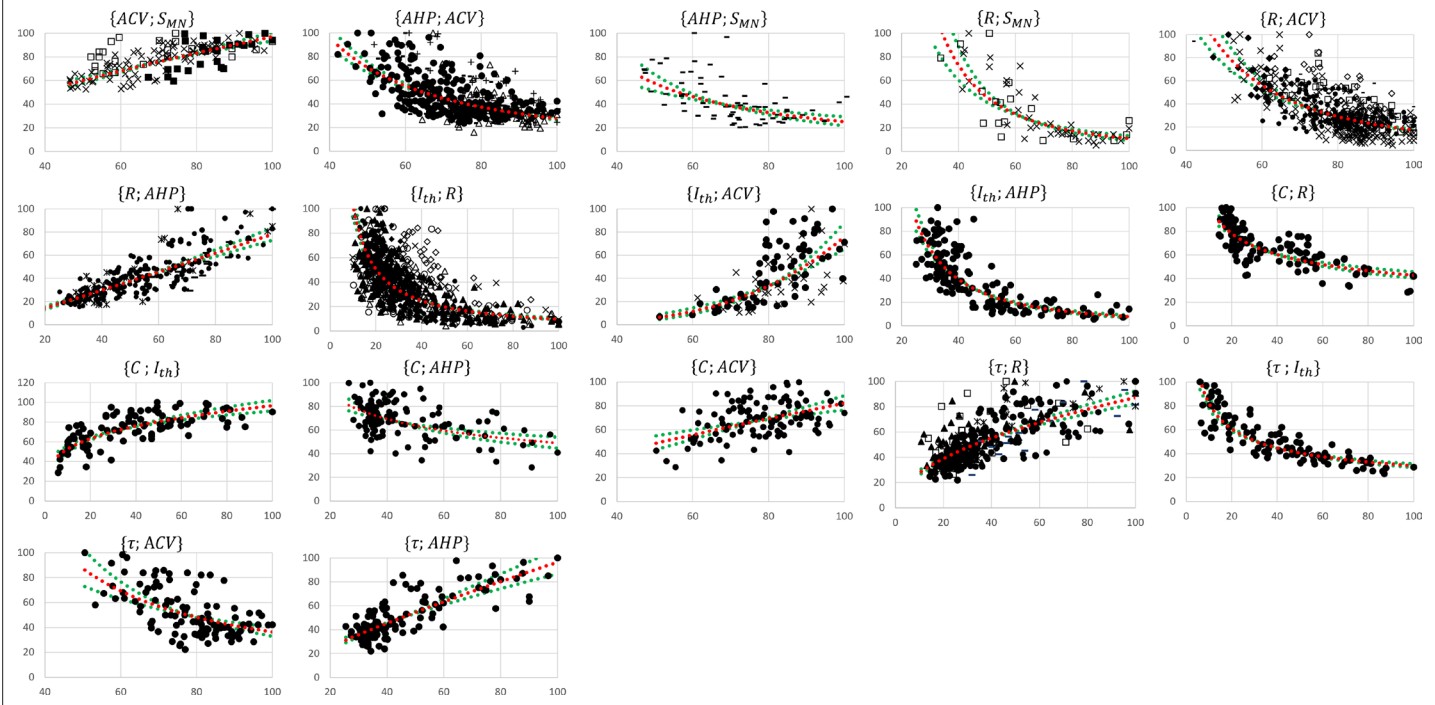

**Figure 3.** Normalized global datasets. These were obtained from the 19 studies reporting cat data that measured and investigated the 17 pairs of motoneuron (MN) properties reported in *Figure 2A*. For each {$A; B$} pair, the property $A$ is read on the y-axis and $B$ on the x-axis. For information, power trendlines $A = k \cdot B^a$ (red dotted curves) are fitted to the data of each dataset and reported in *Table 3*. The 95% confidence interval of the regression is also displayed for each dataset (green dotted lines). The studies are identified with the following symbols: • (*Gustafsson, 1979*; *Gustafsson and Pinter, 1984a*; *Gustafsson and Pinter, 1984b*), ○ (*Munson et al., 1986*), ▲ (*Zengel et al., 1985*), △ (*Foehring et al., 1987*), ■ (*Cullheim, 1978*), □ (*Burke, 1968*; *Burke and ten Bruggencate, 1971*; *Burke et al., 1982*), ◆ (*Krawitz et al., 2001*), ◇ (*Fleshman et al., 1981*), + (*Eccles et al., 1958b*), X (*Kernell, 1966*; *Kernell and Zwaagstra, 1981*; *Kernell and Monster, 1981*), - (*Zwaagstra and Kernell, 1980*), — (*Sasaki, 1991*), ✳ (*Pinter and Vanden Noven, 1989*). The axes are given in % of the maximum retrieved values in the studies consistently with 'Methods' section.

The online version of this article includes the following source data for figure 3:

**Source data 1.** All the cat datasets presented in *Figure 3*.

identified previously, only 1 and 7 experimental datasets were identified to respectively constitute more than 50% and less than 10% of the global dataset they were included in. Therefore, despite a general imbalance towards the over-appearance of the work from *Gustafsson and Pinter, 1984a*; *Gustafsson and Pinter, 1984b* and a tendency towards overlooking some small datasets (*Kernell, 1966*; *Burke and ten Bruggencate, 1971*; *Sasaki, 1991*), the remaining studies all played a significant role in the procedure to constitute the final datasets. Therefore, the inter-study data variability was globally low, and the data distributions reported in the experimental studies were confidently merged into the global datasets plotted in *Figure 3*. It is worth noting that three research groups provided 40, 27, and 15% of the 2717 data points describing the $S_{MN}$, $ACV$, $AHP$, $R$, $I_{th}$, and $\tau$ property distributions in this study, while the $C$-related data was provided by one only research group, leading to potential group-specific methodological bias.

The merged property distributions obtained in the 17 global datasets showed low variability between global datasets, as displayed in *Appendix 1—figure 4*. Indeed, $sd_G < 10\%$ and $\frac{sd_G}{mean_G} < 0.15$ were obtained for all four metrics for all the investigated properties across the global datasets they appeared in. Therefore, the property distributions were similar enough between global datasets to apply the process displayed in *Figure 1* and derive the final size-dependent datasets.

## Size-dependent normalized relationships
The 17 normalized global datasets were processed according to the procedure described in *Figure 1* to derive normalized mathematical relationships between the electrophysiological properties listed in *Table 1* and MN size $S_{MN}$. First, the normalized size-dependent relationship $ACV = 4.0 \cdot S_{MN}^{0.7}$, reported

**Table 3.** Fitted experimental data of pairs of motoneuron (MN) properties and subsequent normalized final size-related relationships. For information, the $r^2$, p-value, and the equation $A = k_a \cdot B^a$ are reported for each fitted global dataset. The normalized MN-size dependent relationships $A = k_c \cdot S_{MN}^c$ are mathematically derived from the transformation of the global datasets and from the power trendline fitting of the final datasets (N data points) as described in 'Methods'. The minimum and maximum values of $k_a$, $k_c$, $a$, and $c$ defining the 95% confidence interval of the regression are also reported in parenthesis for each global and final dataset. The $r^2$ values reported in this table are consistent with the $r^2$ values obtained when directly fitting the normalized experimental datasets with power regressions (see **Appendix 1—table 1**).

| MN property / A | Relationship | $A = k_a \cdot B^a$ (normalized global datasets) | | | | Reference studies | $A = k_c \cdot S_{MN}^c$ (final MN-size dependent datasets) | | | | N points |
|---|---|---|---|---|---|---|---|---|---|---|---|
| | | $k_a$ | $a$ | $r^2$ | p-value | | $k_c$ | $c$ | $r^2$ | p-value | |
| **ACV** | $ACV = k_a \cdot S_{MN}^a$ | 4.0 (2.5; 6.4) | 0.7 (0.6; 0.8) | 0.58 | $<10^{-5}$ | Cullheim, 1978; Kernell and Zwaagstra, 1981; Burke et al., 1982 | 4.0 (2.5; 6.4) | 0.7 (0.6; 0.8) | 0.58 | $<10^{-5}$ | 109 |
| **AHP** | $AHP = k_a \cdot S_{MN}^a$ | $6.1 \cdot 10^3$ ($1.2 \cdot 10^3$; $3.2 \cdot 10^4$) | −1.2 (−1.6; −0.8) | 0.34 | $<10^{-5}$ | Zwaagstra and Kernell, 1980 | | | | | |
| | $AHP = k_a \cdot ACV^a$ | $1.5 \cdot 10^4$ ($7.4 \cdot 10^3$; $2.9 \cdot 10^4$) | −1.4 (−1.5; −1.2) | 0.41 | $<10^{-5}$ | Eccles et al., 1958a, Zwaagstra and Kernell, 1980; Gustafsson and Pinter, 1984b; Foehring et al., 1987 | $2.5 \cdot 10^4$ ($1.2 \cdot 10^4$; $5.0 \cdot 10^4$) | −1.5 (−1.7; −1.3) | 0.41 | $<10^{-5}$ | 492 |
| **R** | $R = k_a \cdot S_{MN}^a$ | $1.5 \cdot 10^5$ ($2.7 \cdot 10^4$; $7.9 \cdot 10^5$) | −2.1 (−2.5; −1.7) | 0.61 | $<10^{-5}$ | Kernell and Zwaagstra, 1981; Burke et al., 1982 | | | | | |
| | $R = k_a \cdot ACV^a$ | $6.3 \cdot 10^5$ ($1.9 \cdot 10^5$; $2.1 \cdot 10^6$) | −2.3 (−2.6; −2.0) | 0.38 | $<10^{-5}$ | Kernell, 1966; Burke, 1968; Barrett and Crill, 1974; Kernell and Zwaagstra, 1981; Fleshman et al., 1981; Gustafsson and Pinter, 1984b; Sasaki, 1991 | $9.6 \cdot 10^5$ ($4.1 \cdot 10^5$; $2.3 \cdot 10^6$) | −2.4 (−2.6; −2.2) | 0.37 | $<10^{-5}$ | 745 |
| | $R = k_a \cdot AHP^a$ | $6.2 \cdot 10^{-1}$ ($4.1 \cdot 10^{-1}$; $9.2 \cdot 10^{-1}$) | 1.1 (0.9; 1.2) | 0.65 | $<10^{-5}$ | Gustafsson, 1979; Gustafsson and Pinter, 1984b; Foehring et al., 1987, Pinter and Vanden Noven, 1989; Sasaki, 1991 | | | | | |
| **$I_{th}$** | $I_{th} = k_a \cdot R^a$ | $1.1 \cdot 10^3$ ($0.8 \cdot 10^3$; $1.3 \cdot 10^3$) | −1.0 (−1.1; −0.9) | 0.37 | $<10^{-5}$ | Kernell, 1966; Fleshman et al., 1981; Gustafsson and Pinter, 1984a; Zengel et al., 1985; Munson et al., 1986; Foehring et al., 1987, Krawitz et al., 2001 | | | | | |
| | $I_{th} = k_a \cdot ACV^a$ | $3.2 \cdot 10^{-6}$ ($1.3 \cdot 10^{-7}$; $8.2 \cdot 10^{-5}$) | 3.7 (3.0; 4.4) | 0.37 | $<10^{-5}$ | Kernell and Monster, 1981; Gustafsson and Pinter, 1984a | $9.0 \cdot 10^{-4}$ ($4.7 \cdot 10^{-4}$; $1.7 \cdot 10^{-3}$) | 2.5 (2.4; 2.7) | 0.37 | $<10^{-5}$ | 722 |
| | $I_{th} = k_a \cdot AHP^a$ | $2.5 \cdot 10^4$ ($1.3 \cdot 10^4$; $4.8 \cdot 10^4$) | −1.7 (−1.9; −1.6) | 0.60 | $<10^{-5}$ | Gustafsson and Pinter, 1984a | | | | | |
| **C** | $C = k_a \cdot R^a$ | $2.4 \cdot 10^2$ ($2.0 \cdot 10^2$; $3.9 \cdot 10^2$) | −0.4 (−0.4; −0.3) | 0.57 | $<10^{-5}$ | Gustafsson and Pinter, 1984b | | | | | |
| | $C = k_a \cdot I_{th}^a$ | $2.9 \cdot 10^1$ ($2.4 \cdot 10^1$; $3.5 \cdot 10^1$) | 0.3 (0.2; 0.3) | 0.51 | $<10^{-5}$ | Gustafsson and Pinter, 1984a | | | | | |
| | $C = k_a \cdot AHP^a$ | $2.8 \cdot 10^2$ ($1.8 \cdot 10^2$; $4.4 \cdot 10^2$) | −0.4 (−0.5; −0.3) | 0.24 | $<10^{-5}$ | Gustafsson and Pinter, 1984b | 1.2 (0.7; 2.0) | 1.0 (0.9; 1.2) | 0.28 | $<10^{-5}$ | 444 |
| | $C = k_a \cdot ACV^a$ | 2.5 (0.7; 8.4) | 0.8 (0.5; 1.0) | 0.17 | $<10^{-5}$ | Gustafsson and Pinter, 1984b | | | | | |
| **$\tau$** | $\tau = k_a \cdot R^a$ | 8.7 (7.2; 10.6) | 0.5 (0.4; 0.6) | 0.52 | $<10^{-5}$ | Burke and ten Bruggencate, 1971; Barrett and Crill, 1974; Gustafsson, 1979; Gustafsson and Pinter, 1984b; Zengel et al., 1985; Pinter and Vanden Noven, 1989; Sasaki, 1991 | | | | | |
| | $\tau = k_a \cdot AHP^a$ | 2.2 (1.3; 3.5) | 0.8 (0.7; 1.0) | 0.63 | $<10^{-5}$ | Gustafsson and Pinter, 1984b | $2.6 \cdot 10^4$ ($1.5 \cdot 10^4$; $4.5 \cdot 10^5$) | −1.5 (−1.6; −1.4) | 0.46 | $<10^{-5}$ | 649 |
| | $\tau = k_a \cdot I_{th}^a$ | $2.3 \cdot 10^2$ ($1.9 \cdot 10^2$; $2.7 \cdot 10^2$) | −0.4 (−0.5; −0.3) | 0.72 | $<10^{-5}$ | Gustafsson and Pinter, 1984a | | | | | |
| | $\tau = k_a \cdot ACV^a$ | $1.2 \cdot 10^4$ ($2.2 \cdot 10^3$; $6.6 \cdot 10^4$) | −1.3 (−1.7; −0.9) | 0.30 | $<10^{-5}$ | Gustafsson and Pinter, 1984b | | | | | |

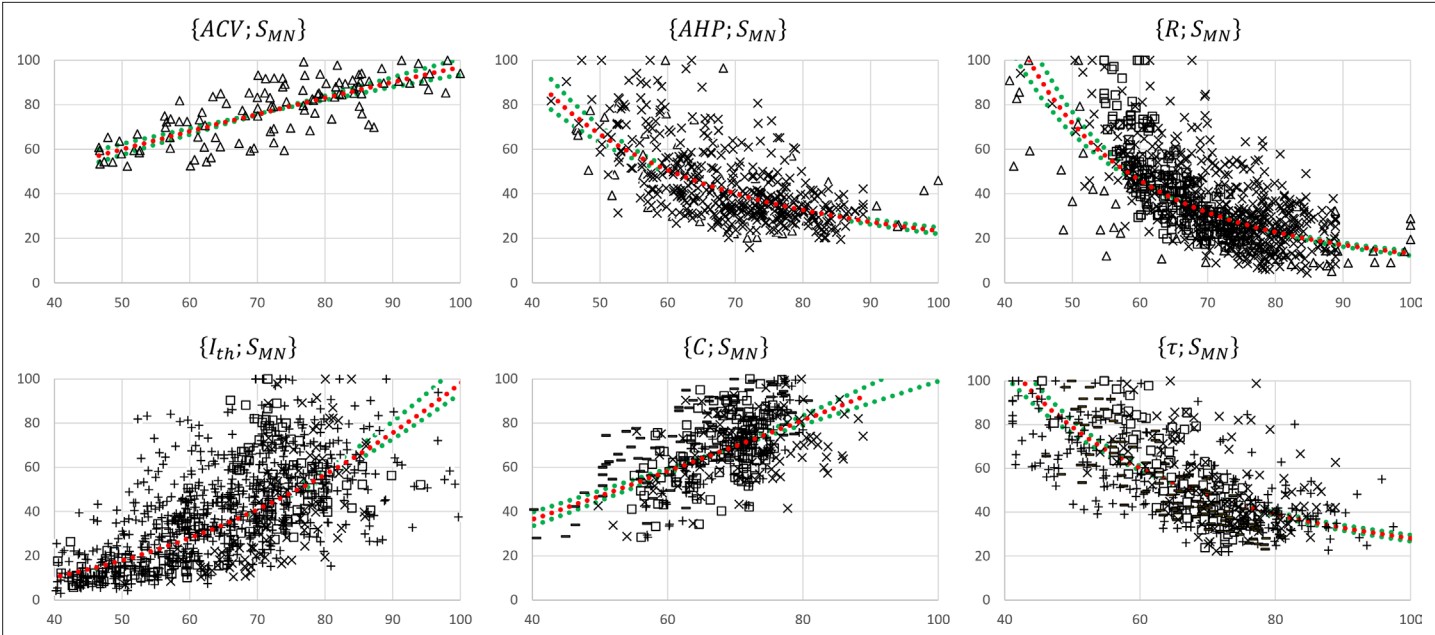

**Figure 4.** Normalized motoneuron (MN) size-related final datasets. These were obtained from the 19 studies reporting cat data that concurrently measured at least two of the morphometric and electrophysiological properties listed in *Table 1*. For each $\{A; S_{MN}\}$ pair, the property $A$ is read on the y-axis and $S_{MN}$ on the x-axis. The power trendlines $A = k_c \cdot S_{MN}^c$ (red dotted curves) are fitted to each dataset and are reported in *Table 3*. The 95% confidence interval of the regression is also displayed for each dataset (green dotted lines). For each $\{A; S_{MN}\}$ plot, the constitutive sub-datasets $\{A; S_{MN}\}$ that were obtained from different global $\{A; B\}$ datasets are specified with the following symbols identifying the property $B$: $S_{MN} \leftrightarrow \triangle$, $ACV \leftrightarrow \times$, $AHP \leftrightarrow \square$, $R \leftrightarrow +$, and $I_{th} \leftrightarrow -$.

in *Table 3*, was derived from the trendline fitting of the normalized $\{ACV; S_{MN}\}$ global dataset, which was obtained from three studies (*Cullheim, 1978*; *Kernell and Zwaagstra, 1981*; *Burke et al., 1982*) and is represented in the upper-left panel in *Figure 3*. This resulted in a statistically significant relationship ($r^2 = 0.58$, p-value<10⁻⁵). A statistically significant ($r^2 = 0.59$, p-value<10⁻⁵) inverse relationship $S_{MN} = k_d \cdot ACV^d$ was also fitted to this dataset. Then, the $\{AHP; ACV\}$ dataset, represented in the second panel upper row in *Figure 3* and obtained from four studies (*Eccles et al., 1958a*; *Zwaagstra and Kernell, 1980*; *Gustafsson and Pinter, 1984b*; *Foehring et al., 1987*), was transformed into a new $\{AHP; S_{MN}\}_1$ dataset by applying the priory-derived $S_{MN} = k_d \cdot ACV^d$ relationship to the list of $ACV$ values. This transformed $\{AHP; S_{MN}\}_1$ dataset was then merged with the $\{AHP; S_{MN}\}_2$ dataset (third panel upper row in *Figure 2*) obtained from *Zwaagstra and Kernell, 1980*, yielding the final $\{AHP; S_{MN}\}_f$ dataset of $N = 492$ data points shown in *Figure 4*, second panel. A statistically significant ($r^2 = 0.58$, p-value<10⁻⁵) power trendline $AHP = 2.5 \cdot 10^4 \cdot S_{MN}^{-1.5}$ was fitted to the $\{AHP; S_{MN}\}_f$ dataset and is reported in *Table 3*. As before, a statistically significant ($r^2 = 0.38$, p-value<10⁻⁵) inverse relationship $S_{MN} = k_d \cdot AHP^d$ was also fitted to this dataset for future use. A similar procedure was applied to derive the normalized final relationships between $\{R, I_{th}, C, \tau\}$ and $S_{MN}$ reported in the last column of *Table 3*. A statistically significant (p-value<10⁻⁵) correlation was obtained between each electrophysiological property $\{ACV, AHP, R, I_{th}, C, \tau\}$ and $S_{MN}$ as reported in *Table 3*. With $r^2 \in [0.28; 0.58]$, it was obtained that $\{I_{th}, C, ACV\}$ and $\{R, AHP, \tau\}$ respectively increased and decreased with increasing MN sizes $S_{MN}$, with slopes reported in *Table 3* and plotted in *Figure 5*.

## Normalized mathematical relationships between electrophysiological properties

The final size-dependent relationships reported in *Table 3* were mathematically combined to yield normalized relationships between all electrophysiological properties listed in *Table 1*. As an example, $I_{th} = 9.4 \cdot 10^{-4} \cdot S_{MN}^{2.5}$ and $R = 9.6 \cdot 10^5 \cdot S_{MN}^{-2.4}$ were obtained in *Table 3*. When mathematically combined (with the latter mathematically inverted as $S_{MN} = 2.9 \cdot 10^2 \cdot R^{-0.4}$), these relationships yielded the normalized relationship between rheobase and input resistance $I_{th} = 1.5 \cdot 10^3 \cdot R^{-1}$. All

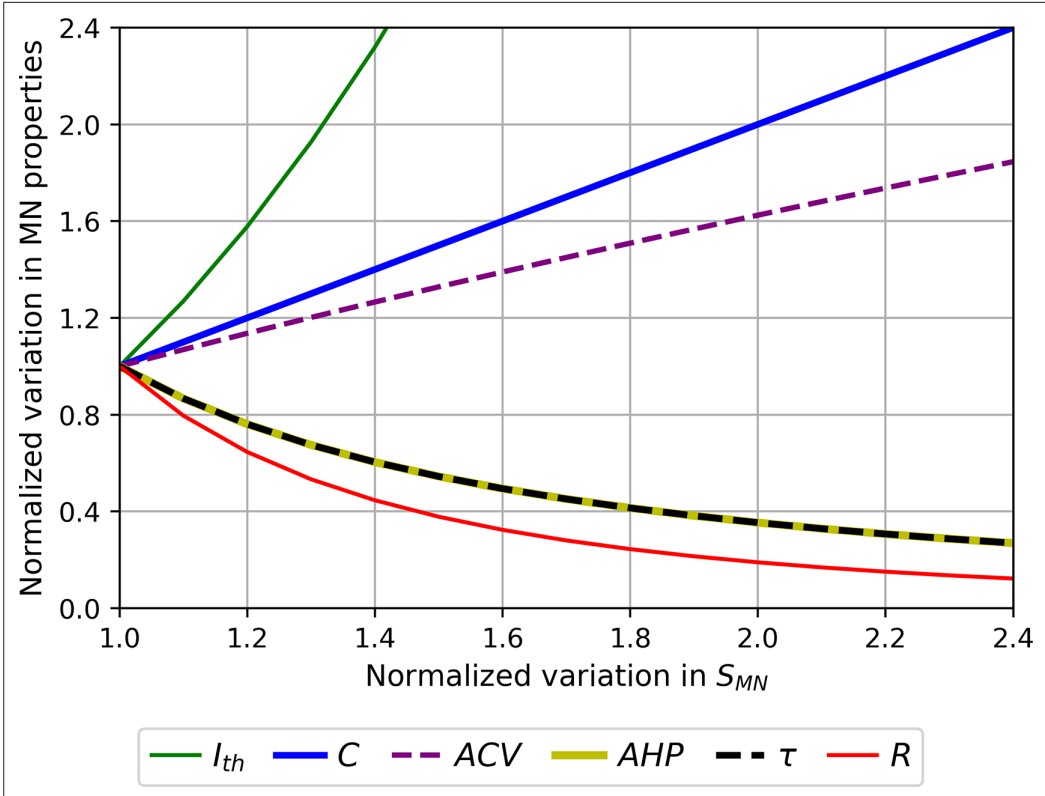

**Figure 5.** Normalized size-dependent behaviour of the motoneuron (MN) properties $ACV$, $AHP$, $R$, $I_{th}$, $C$, and $\tau$. For displaying purposes, the MN properties are plotted in arbitrary units as power functions (intercept $k = 1$) of $S_{MN}$: $A = S_{MN}^c$ according to **Table 3**. The larger the MN size, the larger $ACV$, $C$, and $I_{th}$ in the order of increasing slopes, and the lower $AHP$, $\tau$ and $R$ in the order of increasing slopes.

normalized relationships between MN electrophysiological properties hence obtained are provided in **Appendix 1—table 2**.

## Validation of the normalized relationships

The normalized relationships reported in **Appendix 1—table 2** were obtained from the complete global datasets represented in **Figure 3**. A fivefold cross-validation of these relationships was performed, as described in 'Methods', and assessed with calculations of the $nME$, $nRMSE$, and $r_{pred}^2$ validation metrics averaged across the five permutations. As displayed in **Figure 6**, the normalized relationships obtained from the 17 training datasets predicted, in average across the five permutations, the experimental data from the test datasets with average $nME$ between 52 and 300%, $nRMSE$ between 13 and 24%, and coefficients of determination $r_{pred}^2$ between 0.20 and 0.74 and of the same order of magnitude as the experimental $r_{exp}^2$ values.

## Scaling of the normalized relationships

The normalized relationships provided in **Appendix 1—table 2** were finally scaled to typical cat data obtained from the literature following the procedure described in 'Methods'. $D_{soma}$ and $S_{neuron}$ were found to vary in cats over an average $q_S^E = 2.4$-fold range according to a review of the morphometric studies reported in the two first lines of **Appendix 1—table 3**. $q_S^E$ may however be underestimated for reasons discussed in Appendix 1. Then, the empirical $q_A^E$ and theoretical $q_A^T$ ratios, defined in 'Methods', were calculated for each MN electrophysiological property $A$ and are reported in **Appendix 1—table 4**. For example, MN resistance $R$ was found to vary over an average $q_R^E = 10.9$-fold range in an MN pool according to the literature, while the theoretical fold range $q_R^T = \left( q_S^E \right)^{|c_R|} = 2.4^{2.4} = 8.4$ was obtained from the $R = 9.6 \cdot 10^5 \cdot S_{MN}^{-2.4}$ normalized relationship previously derived when a 2.4-fold range is set

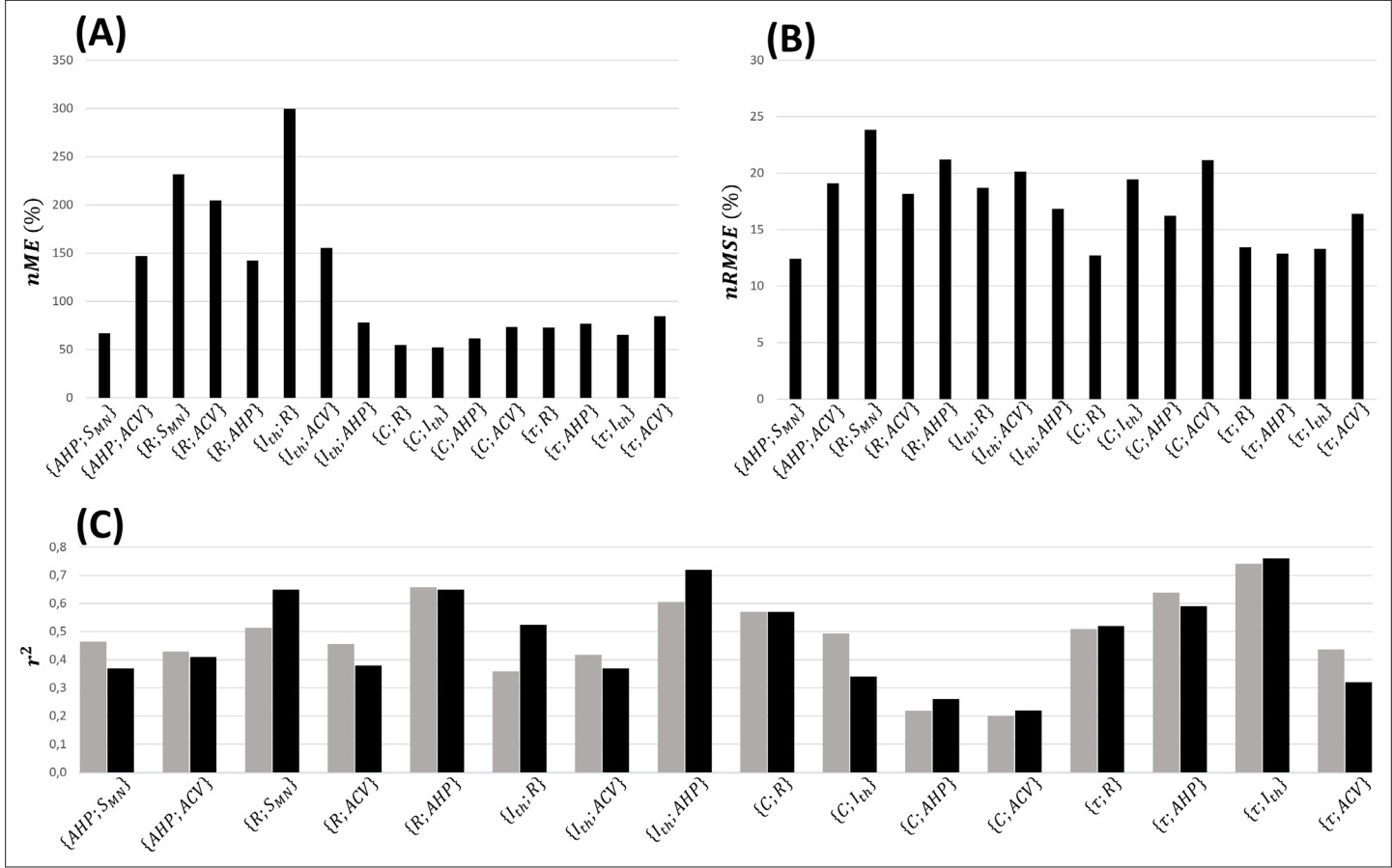

**Figure 6.** Fivefold cross-validation of the normalized mathematical relationships. Here are reported for each dataset the average values across the five permutations of (**A**) the normalized maximum error (*nME*), (**B**) the normalized root mean square error (*nRMSE*), and (**C**) coefficient of determination ($r^2_{pred}$, grey bars), which is compared with the coefficient of determination ($r^2_{exp}$, black bars) of the power trendline fitted to the log–log transformation of global experimental datasets directly.

for $S_{MN}$. As shown in *Appendix 1—table 4*, $\frac{q^E_A}{q^T_A} \in [0.8; 1.3]$ for the MN properties *ACV*, *AHP*, *C*, and $\tau$, while the theoretical ranges for $R$ and $I_{th}$ span over a narrower domain than expected from experimental results ($q^T_R = 8.4 \ll 10.9 = q^E_R$ and $q^T_{I_{th}} = 9.1 \ll 16.3 = q^E_{I_{th}}$). This suggests that the range of variation of $R$ and $I_{th}$ is not entirely explained by the variation in MN size $S_{MN}$, and that MN excitability is not only determined by $S_{MN}$, as reviewed in *Powers and Binder, 2001*.

The normalized relationships were finally scaled using the theoretical ranges reported in the last column of *Appendix 1—table 4*. Taking the $\{I_{th}; R\}$ pair as example, as $I_{th} = 1.4 \cdot 10^3 \cdot R^{-1}$ (normalized relationship derived previously and provided in *Appendix 1—table 2*), $R \in [0.5; 4.0] \cdot 10^6 \Omega$ and $I_{th} \in [3.9; 35.0] \cdot 10^{-9} A$ (*Appendix 1—table 4*), it was directly obtained from 'Methods' that $I_{th} = \frac{2.7 \cdot 10^{-2}}{R}$ in SI base units. A similar approach yielded the final mathematical relationships reported in *Table 4* between all MN electrophysiological and morphometric properties *ACV*, *AHP*, *R*, $I_{th}$, *C*, $\tau$, and $S_{MN}$. All constants and relationships are given in SI base units.

## Predicting correlations between MN properties

With this approach, the correlations between MN properties that were never concurrently measured in the literature are predicted, as displayed in *Figure 2B*. Such unknown relationships were indirectly extracted from the combination of known relationships (*Table 3*) and typical ranges of values obtained from the literature for these properties. Due to the prior fivefold cross-validation of the relationships in *Table 4*, these findings are reliable as indirectly consistent with the literature data processed in this study.

**Table 4.** Mathematical empirical cat relationships between the motoneuron (MN) properties $S_{neuron}$, $D_{soma}$, $R$, $R_m$, $C$, $\tau$, $I_{th}$, $AHP$, and $ACV$. Each column provides the relationships between one and the eight other MN properties. If one property is known, the complete MN profile can be reconstructed by using the pertinent line in this table. All constants and properties are provided in SI base units (metres, seconds, ohms, farads, and amperes). The relationships involving $R_m$ were obtained from theoretical relationships involved in Rall's cable theory (see 'Discussion').

| | $S_{neuron}$ [m²] | $D_{soma}$ [m] | $R$ [Ω] | $R_m$ [Ω·m²] | $C$ [F] | $\tau$ [s] | $I_{th}$ [A] | $AHP$ [s] | $ACV$ [m·s⁻¹] |
|---|---|---|---|---|---|---|---|---|---|
| $S_{neuron}$ [m²] | | $S_{neuron} = 5.5 \cdot 10^{-3} \cdot D_{soma}$ | $S_{neuron} = \dfrac{9.5 \cdot 10^{-5}}{R^{0.41}}$ | $S_{neuron} = \dfrac{1.5 \cdot 10^{-7}}{R_m^{0.70}}$ | $S_{neuron} = 1.2 \cdot 10^{2} \cdot C$ | $S_{neuron} = \dfrac{8.4 \cdot 10^{-9}}{\tau^{0.67}}$ | $S_{neuron} = 4.0 \cdot 10^{-4} \cdot I_{th}^{0.40}$ | $S_{neuron} = \dfrac{5.5 \cdot 10^{-8}}{AHP^{0.66}}$ | $S_{neuron} = 4.6 \cdot 10^{-10} \cdot ACV^{1.44}$ |
| $D_{soma}$ [m] | $D_{soma} = 1.8 \cdot 10^{2} \cdot S_{neuron}$ | | $D_{soma} = \dfrac{1.7 \cdot 10^{-2}}{R^{0.41}}$ | $D_{soma} = \dfrac{2.6 \cdot 10^{-5}}{R_m^{0.70}}$ | $D_{soma} = 2.1 \cdot 10^{4} \cdot C$ | $D_{soma} = \dfrac{1.5 \cdot 10^{-6}}{\tau^{0.67}}$ | $D_{soma} = 7.1 \cdot 10^{-2} \cdot I_{th}^{0.40}$ | $D_{soma} = \dfrac{9.8 \cdot 10^{-6}}{AHP^{0.66}}$ | $D_{soma} = 8.3 \cdot 10^{-8} \cdot ACV^{1.44}$ |
| $R$ [Ω] | $R = \dfrac{1.7 \cdot 10^{-10}}{S_{neuron}^{2.43}}$ | $R = \dfrac{5.1 \cdot 10^{-5}}{D_{soma}^{2.43}}$ | | $R = 6.8 \cdot 10^{6} \cdot R_m^{1.70}$ | $R = \dfrac{1.5 \cdot 10^{-15}}{C^{2.48}}$ | $R = 7.1 \cdot 10^{9} \cdot \tau^{1.64}$ | $R = \dfrac{3.1 \cdot 10^{-2}}{I_{th}^{0.96}}$ | $R = 7.5 \cdot 10^{7} \cdot AHP^{1.61}$ | $R = \dfrac{8.3 \cdot 10^{12}}{ACV^{3.50}}$ |
| $R_m$ [Ω·m²] | $R_m = \dfrac{1.0 \cdot 10^{-10}}{S_{neuron}^{1.43}}$ | $R_m = \dfrac{1.7 \cdot 10^{-7}}{D_{soma}^{1.43}}$ | $R_m = 5.7 \cdot 10^{-5} \cdot R^{0.59}$ | | $R_m = \dfrac{1.7 \cdot 10^{-13}}{C^{1.46}}$ | $R_m = 3.6 \cdot 10^{1} \cdot \tau^{0.96}$ | $R_m = \dfrac{7.4 \cdot 10^{-6}}{I_{th}^{0.57}}$ | $R_m = 2.5 \cdot AHP^{0.95}$ | $R_m = \dfrac{2.3 \cdot 10^{3}}{ACV^{2.06}}$ |
| $C$ [F] | $C = 1.3 \cdot 10^{-2} \cdot S_{neuron}$ | $C = 7.9 \cdot 10^{-5} \cdot D_{soma}$ | $C = \dfrac{1.5 \cdot 10^{-6}}{R^{0.40}}$ | $C = \dfrac{1.8 \cdot 10^{-9}}{R_m^{0.69}}$ | | $C = \dfrac{1.6 \cdot 10^{-10}}{\tau^{0.66}}$ | $C = 5.9 \cdot 10^{-6} \cdot I_{th}^{0.39}$ | $C = \dfrac{9.7 \cdot 10^{-10}}{AHP^{0.65}}$ | $C = 9.0 \cdot 10^{-12} \cdot ACV^{1.41}$ |
| $\tau$ [s] | $\tau = \dfrac{1.0 \cdot 10^{-12}}{S_{neuron}^{1.48}}$ | $\tau = \dfrac{2.3 \cdot 10^{-9}}{D_{soma}^{1.48}}$ | $\tau = 9.6 \cdot 10^{-7} \cdot R^{0.61}$ | $\tau = 1.4 \cdot 10^{-2} \cdot R_m^{1.04}$ | $\tau = \dfrac{8.6 \cdot 10^{-16}}{C^{1.51}}$ | | $\tau = \dfrac{1.2 \cdot 10^{-7}}{I_{th}^{0.59}}$ | $\tau = 6.2 \cdot 10^{-2} \cdot AHP^{0.98}$ | $\tau = \dfrac{7.4 \cdot 10^{1}}{ACV^{2.14}}$ |
| $I_{th}$ [A] | $I_{th} = 3.8 \cdot 10^{8} \cdot S_{neuron}^{2.52}$ | $I_{th} = 7.8 \cdot 10^{2} \cdot D_{soma}^{2.52}$ | $I_{th} = \dfrac{2.7 \cdot 10^{-2}}{R^{1.04}}$ | $I_{th} = \dfrac{2.2 \cdot 10^{-9}}{R_m^{1.76}}$ | $I_{th} = 6.5 \cdot 10^{13} \cdot C^{2.57}$ | $I_{th} = \dfrac{1.6 \cdot 10^{-12}}{\tau^{1.70}}$ | | $I_{th} = \dfrac{1.8 \cdot 10^{-10}}{AHP^{1.67}}$ | $I_{th} = 1.1 \cdot 10^{-15} \cdot ACV^{3.64}$ |
| $AHP$ [s] | $AHP = \dfrac{1.0 \cdot 10^{-11}}{S_{neuron}^{1.51}}$ | $AHP = \dfrac{2.7 \cdot 10^{-8}}{D_{soma}^{1.51}}$ | $AHP = 1.3 \cdot 10^{-5} \cdot R^{0.62}$ | $AHP = 2.2 \cdot 10^{-1} \cdot R_m^{1.06}$ | $AHP = \dfrac{7.7 \cdot 10^{-15}}{C^{1.54}}$ | $AHP = 1.7 \cdot 10^{1} \cdot \tau^{1.02}$ | $AHP = \dfrac{1.5 \cdot 10^{-6}}{I_{th}^{0.60}}$ | | $AHP = \dfrac{1.4 \cdot 10^{3}}{ACV^{2.18}}$ |
| $ACV$ [m·s⁻¹] | $ACV = 3.0 \cdot 10^{6} \cdot S_{neuron}^{0.69}$ | $ACV = 8.1 \cdot 10^{4} \cdot D_{soma}^{0.69}$ | $ACV = \dfrac{4.9 \cdot 10^{3}}{R^{0.29}}$ | $ACV = \dfrac{5.4 \cdot 10^{1}}{R_m^{0.48}}$ | $ACV = 8.2 \cdot 10^{7} \cdot C^{0.71}$ | $ACV = \dfrac{7.5}{\tau^{0.47}}$ | $ACV = 1.3 \cdot 10^{4} \cdot I_{th}^{0.27}$ | $ACV = \dfrac{2.7 \cdot 10^{1}}{AHP^{0.46}}$ | |

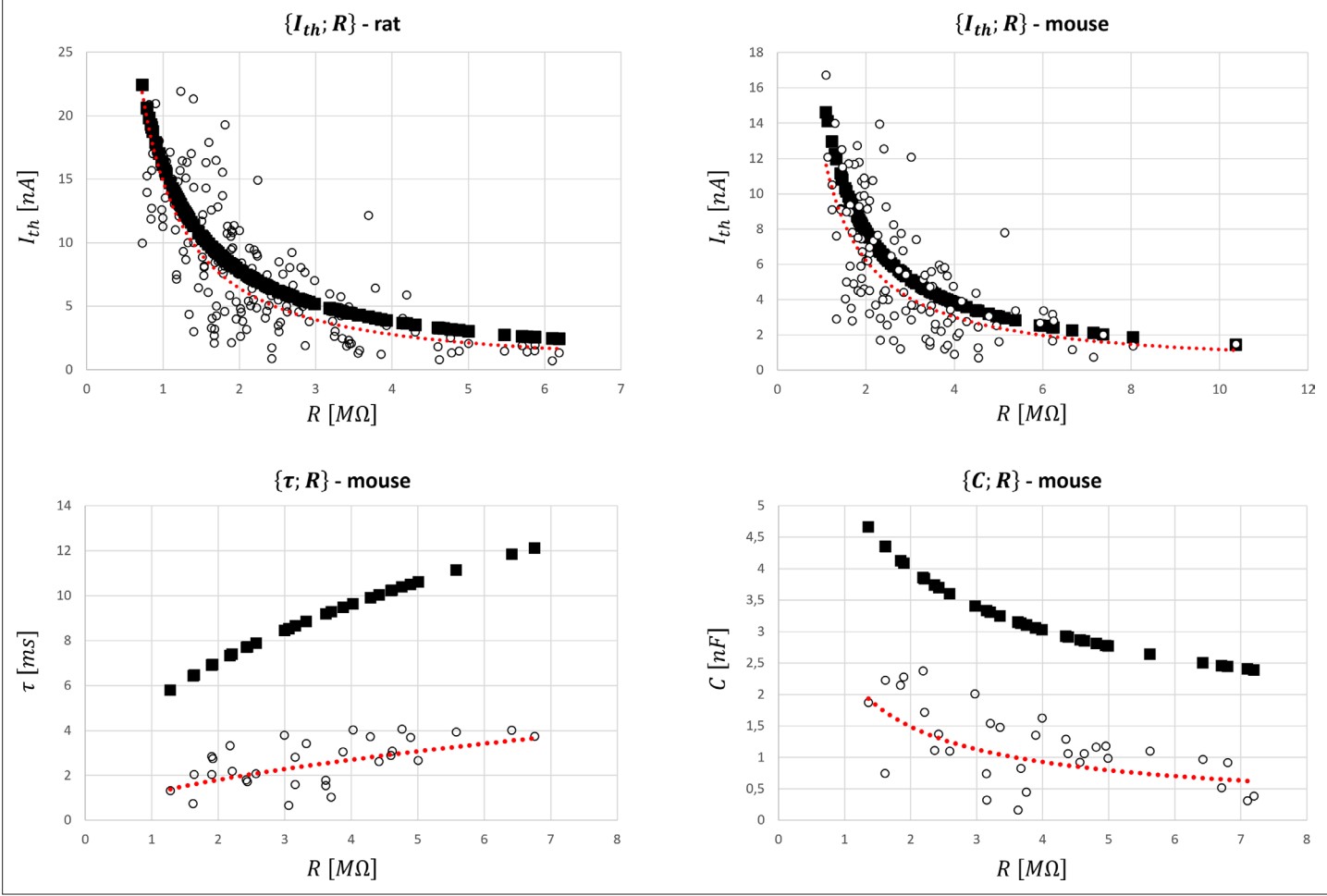

**Figure 7.** Global datasets for rat and mouse species and predictions of the motoneuron (MN) properties with the cat mathematical relationships (**Table 4**). They were obtained from the five studies reporting data on rats and the four studies presenting data on mice reported in **Appendix 1—table 5** that measured the $\{I_{th}; R\}$, $\{\tau; R\}$, and $\{C; R\}$ pairs of MN electrophysiological properties. $I_{th}$, $R$, $\tau$, $C$ are given in nA, MΩ, ms, and nF, respectively. The experimental data (O symbol) is fitted with a power trendline (red dotted curve) and compared to the predicted quantities obtained with the scaled cat relationships in **Table 4** (■ symbol).

The online version of this article includes the following source data for figure 7:

**Source data 1.** Datasets used in the rat plot.

**Source data 2.** Datasets used in the mouse plots.

### Extrapolation to rats and mice

It was assessed whether the scaled relationships obtained from cat data and reported in **Table 4** could be successfully applied to rat and mouse data. A global $\{I_{th}; R\}$ dataset was obtained from five studies focussing on rats reported in **Appendix 1—table 5**, while four mice studies reported in **Appendix 1—table 5** provided processable data for the pairs $\{I_{th}; R\}$, $\{\tau; R\}$, and $\{C; R\}$. Only data from control groups of wild-type animals were considered while data on mutated, operated, or trained animals were disregarded. Because the information on the age distribution of the retrieved datasets is not available, data from adults rats and mice of unknown age were used, despite well-known age-related variations in the MN electrophysiological properties in these animals (**Highlander et al., 2020**; **Huh et al., 2021**). Like the cat data, low variability was observed between the $R$ and $I_{th}$ distributions reported by different studies in the rat and mouse $\{I_{th}; R\}$ datasets. **Figure 7** reports the global datasets for all four pairs of properties and the predictions obtained with the cat relationships reported in **Table 4**. As displayed in **Appendix 1—table 5**, $r^2_{pred}$ and $r^2_{exp}$ values were close for all four datasets, with a maximum 12.5% difference for $\{\tau; R\}$ in mice. $nRMSE$ values remained low in both rat and mouse $\{I_{th}; R\}$ datasets, while being substantially higher in both $\{\tau; R\}$ and $\{C; R\}$ datasets in mice because of an inter-species-specific offset in the relationships.

**Table 5.** Fitted experimental data of pairs of one muscle unit (mU) and one motoneuron (MN) property and subsequent $S_{mU} \propto S_{MN}^c$ relationships.

The $r^2$, p-value, and the equation $A \propto B^c$ of the trendlines are reported for each pair of properties. In the last column, prior knowledge is used to derive relationships.

| | | $A = k \cdot B^c$ (fitted relationships) | | | | | (final relationships) |
|---|---|---|---|---|---|---|---|
| Species | Relationship | $c$ | $r^2$ | p-value | Reference studies | | $c$ |
| Rat | $IR = k \cdot ACV^c$ | 3.4 | 0.45 | $6.10^{-3}$ | *Kanda and Hashizume, 1992* | | 2.4 |
| | $F^{tw} = k \cdot ACV^c$ | 9.4 | 0.43 | $<10^{-5}$ | *Knott et al., 1971* | | 6.6 |
| | $F^{tet} = k \cdot ACV^c$ | 7.2 | 0.37 | $<10^{-5}$ | *Mcphedran et al., 1965; Wuerker et al., 1965; Appelberg and Emonet-Dénand, 1967; Proske and Waite, 1974; Bagust, 1974; Jami and Petit, 1975; Stephens and Stuart, 1975; Burke et al., 1982; Emonet-Dénand et al., 1988* | | 5.0 |
| | $F^{tet} = k \cdot R^c$ | -1.3 | 0.27 | $6.10^{-5}$ | *Dum and Kennedy, 1980* | | 3.2 |
| Cat | $F^{tet} = k \cdot S_{MN}^c$ | 2.0 | 0.21 | $2.10^{-2}$ | *Burke et al., 1982* | | 2.0 |
| | $F^{tw} = k \cdot R^c$ | -2.1 | 0.42 | $<10^{-5}$ | *Manuel and Heckman, 2011; Martínez-Silva et al., 2018* | | 5.1 |
| | $F^{tw} = k \cdot I_{th}^c$ | 1.3 | 0.64 | $2.10^{-4}$ | *Manuel and Heckman, 2011* | | 3.3 |
| Mouse | $F^{tet} = k \cdot I_{th}^c$ | 1.0 | 0.80 | $6.10^{-2}$ | *Manuel and Heckman, 2011* | | 2.5 |
| Mean ± sd | | | | | | | 3.8 ± 1.5 |

The online version of this article includes the following source data for table 5:

**Source data 1.** Numerical data used to derive the relationships presented in *Table 5*.

## Relationships between MN and mU properties

As shown in *Figure 2A*, eight pairs of one MN and one mU property were investigated in 14 studies in the literature in cats and rats. As remarked by *Heckman and Enoka, 2012* and *Manuel et al., 2019*, most recent studies were performed on mice, while most dated were dominantly performed on cats. One study on the rat gastrocnemius muscle (*Kanda and Hashizume, 1992*) indicated no correlation between $IR$ and $ACV$. However, after removing from the dataset two outliers that fell outside 2 standard deviations of the mean $IR$ data, a statistically significant correlation (p<0.05) between $IR$ and $ACV$ was successfully fitted with a power trendline ($r^2 = 0.43$). Eight studies, dominantly focussing on the cat soleus and medial gastrocnemius muscles, found a strong correlation between $F^{tet}$ and $ACV$, while one study showed a significant correlation for the pair $\{F^{tet}; R\}$ in both the cat tibialis anterior and extensor digitorum longus muscles. One study (*Burke et al., 1982*) on the cat soleus, medial and lateral gastrocnemius muscles inferred a statistically significant correlation for $\{F^{tet}; S_{MN}\}$. In mice, both $R$ and $I_{th}$ were significantly related to muscle tetanic and twitch forces. As $F^{tet}$, $F^{tw}$, and $IR$ are reliable indices of $S_{mU}$ as discussed in 'Methods, the MN size-dependent relationships in *Table 3* return eight $S_{mU} = k \cdot S_{MN}^c$ relationships between mU and MN properties (*Table 5*). A 2.8-fold range in positive $c$-values $c \in [2.4; 6.6]$ (mean $3.8 \pm 1.5$ sd) was obtained between studies. This result infers that mU and MN size are positively related in cats, mice, and potentially in rats.

## Discussion

We processed the normalized data (*Figure 3*) from previous cat experimental studies to extract mathematical relationships (*Table 4*) between several MN electrophysiological and morphological properties (*Table 1*), providing a clear summary of previously known qualitative inter-relations between these MN properties. *Table 4* is a new convenient tool for neuroscientists, experimenters, and modellers. Besides obtaining significant quantitative correlations between MN electrophysiological properties and MN size (*Table 3*, *Figure 4*, *Figure 5*), we established (*Table 4*) and validated (*Figure 6*) a comprehensive mathematical framework that models an extensive set of experimental datasets available in the literature, quantitatively links all the pairs of the MN properties in *Table 1*, and is consistent with Rall's cable theory, as discussed in the following. This framework is directly applicable, although with limitations, to other mammalian species (*Figure 7*), and is used along with other measurements of mU properties (*Table 5*) to discuss the Henneman's size principle.

## Consistency of the relationships with previous empirical results and Rall's theory

The mathematical relationships derived in *Table 4* are first in strong agreement with the relative variations between MN properties that were reported in review papers and are listed in 'Introduction'. Some combinations of the relationships in *Table 4* are besides consistent with other cat experimental data that were not included in the global datasets and were not used for deriving the relationships. For example, the relationships $I^{th} = k \cdot \tau^{-1.7}$, $\frac{1}{C} = k \cdot \tau^{0.7}$, $\frac{1}{S_{MN}} = k \cdot ACV^{-1.4}$, and $\frac{1}{R} = k \cdot ACV^{3.5}$ in *Table 4* yield, when combined, $\frac{I^{th}}{C} = k \cdot \tau^{-1}$ and $\frac{1}{R \cdot S_{MN}} = k \cdot ACV^{2.1}$, which are consistent with the relationship $\frac{I^{th}}{C} = k \cdot \tau^{-1}$ experimentally observed by *Gustafsson and Pinter, 1985* and $\frac{1}{R \cdot S_{MN}} = k \cdot ACV^{1.9}$ measured by *Kernell and Zwaagstra, 1981*. The stronger-than-linear inverse relationship $R = k \cdot S_{MN}^{-2.4}$ is also consistent with the phenomenological conclusions by *Kernell and Zwaagstra, 1981*. Moreover, some standard equations resulting from Rall's cable theory can be derived with a combination of the relationships in *Table 4*, assuming the MNs as equivalent cylinders with spatial uniformity of both $C_m$ and $R_m$. For example, taking $\frac{\tanh(L)}{L} = 0.5$, which is in the typical $[0.48; 0.76]$ range reported in the literature (*Powers and Binder, 2001*), the relation $C = \frac{\tau}{R} \cdot \frac{L}{\tanh(L)}$ (eq. 1. in *Gustafsson and Pinter, 1984a*) is obtained when combining the $R - S_{neuron}$, $C - S_{neuron}$, and $\tau - S_{neuron}$ relationships. Also, taking $\frac{\tanh(L)}{L} = 0.6$ (standard value provided in *Powers and Binder, 2001*), the $R - S_{neuron}$ equation in *Table 4* applied to $R = \frac{R_m}{S_{neuron}} \cdot \frac{L}{\tanh(L)}$ in *Rall, 1977*, yields $R_m = \frac{1.0 \cdot 10^{-10}}{S_{neuron}^{1.4}}$. This $R_m - S_{neuron}$ relationship is consistent with the observations in *Powers and Binder, 2001* that smaller MNs have larger $R_m$ values. Also, considering that $q_S^E = 2.4$ and $S_{neuron} \in [0.18; 0.44]$ $mm^2$ (*Appendix 1—table 3*), $R_m = \frac{1.0 \cdot 10^{-10}}{S_{MN}^{1.4}}$ makes $R_m$ vary over the 3.5-fold theoretical range $[0.12; 0.44]$ $\Omega \cdot m^2$ in the MN pool. This is highly consistent both with the $q_A^E = 3.2$-fold range and the $[A_{min}^E; A_{max}^E] = [0.16; 0.59]$ $\Omega \cdot m^2$ mean range previously reported in the literature (*Albuquerque and Thesleff, 1968*; *Barrett and Crill, 1974*; *Burke et al., 1982*; *Gustafsson and Pinter, 1984b*). Also, $R_m = \frac{1.0 \cdot 10^{-10}}{S_{neuron}^{1.4}}$ yields $R_m = 2.5 \cdot AHP^{0.9}$ from the relationships in *Table 4*, which is consistent with the indirect conclusion of a positive correlation between $R_m$ and $AHP$ in *Gustafsson and Pinter, 1984b*. Finally, $R = \frac{0.6 R_m}{S_{neuron}}$ and $R_m = \frac{1.0 \cdot 10^{-10}}{S_{neuron}^{1.4}}$ support the statement that the variations in $R_m$ in the MN pool may be as important as cell size $S_{neuron}$ in determining MN excitability (*Gustafsson and Pinter, 1985*), although the variation of $R$ in an MN pool may be also related to cell geometry (*Gustafsson and Pinter, 1984b*), which was not investigated in this study. This is consistent with the results in *Appendix 1—table 4*, which demonstrate that the range of variation $q_S^E$ of $S_{MN}$ does not entirely explain the ranges of variation $q_R^E$ and $q_{I_{th}}^E$ of $R$ and $I_{th}$, and that MN excitability is not only determined by MN size. Because of the aforementioned consistency with previous findings and with Rall's theory, the remaining relationships between $R_m$ and the other MN properties were calculated from $R_m = \frac{1.0 \cdot 10^{-10}}{S_{neuron}^{1.4}}$, following the same methods as before, and added to *Table 4*. Importantly, $R_m$ might however not be uniform across the somatodendritic membrane, according to results from completely reconstructed MNs (*Fleshman et al., 1988*), and might be positively correlated to the somatofugal distance (*Fleshman et al., 1983*), with dendritic $R_m$ being higher than somatic $R_m$ (*Barrett and Crill, 1974*; *Powers and Binder, 2001*; *Olinder et al., 2006*) by a factor 100–300 (*Clements and Redman, 1989*). This contradicts the assumptions of membrane isopotentiality necessary to apply Rall's cable theory. The relationships in *Table 4*, which were obtained from measurements in the soma where $R_m$ is mainly constant, may thus not directly extrapolate to the dendritic regions of the MNs. Yet, the dimensionless variations of the average across MN surface of $R_m$ with the other MN properties may still be valid (*Olinder et al., 2006*).

The equations in *Table 4* support the notion that MNs behave like resistive-capacitive systems, as demonstrated by the combination of the $R - \tau$ and $C - \tau$ relationships in *Table 4*, which yields $\tau^{0.9} = 0.8 \cdot RC$, close to the theoretical equation $\tau = RC \cdot \frac{\tanh(L)}{L}$ derived from Rall's theory. Also, *Table 4* directly yields, from the combination of $R_m = \frac{1.0 \cdot 10^{-10}}{S_{neuron}^{1.4}}$ with the other relationships, $\tau = 1.4 \cdot 10^{-2} \cdot R_m$, which is exactly eq. 2.15 in *Rall, 1977* ($\tau = R_m C_m$) with a constant value for $C_m = 1.4 \cdot 10^{-2} F \cdot m^{-2}$ that is consistent with the literature.

The relationship $C = 1.3 \cdot 10^{-2} \cdot S_{neuron}$ obtained in *Figure 4* and reported in *Table 4* is in agreement with the widely accepted proportional relationship between MN membrane capacitance and surface area, when spatial uniform values of $R_m$ and $C_m$ are assumed. It is worth noting that this proportional relationship was obtained in this study without simultaneous empirical measurements

of $S_{MN}$ and $C$. The values of $C$ were obtained from Rall's $C = \frac{\tau}{R} \cdot \frac{L}{\tanh(L)}$ equation (**Gustafsson and Pinter, 1984b**) and the corresponding values of $S_{MN}$ were indirectly obtained from the final datasets $\{ACV; S_{MN}\}$, $\{AHP; S_{MN}\}$, $\{R; S_{MN}\}$ and $\{I_{th}; S_{MN}\}$ that involved the data from 17 selected studies as described in **Figure 1**. Although consistent with typical values reported in the literature (**Lux and Pollen, 1966**; **Albuquerque and Thesleff, 1968**; **Barrett and Crill, 1974**; **Adrian and Hodgkin, 1975**; **Sukhorukov et al., 1993**; **Major et al., 1994**; **Solsona et al., 1998**; **Thurbon et al., 1998**; **Gentet et al., 2000**), the $C_m = 1.3 \cdot 10^{-2} F \cdot m^{-2}$ constant value reported in this study is slightly higher than the widely accepted $C_m = 1.0 \cdot 10^{-2} F \cdot m^{-2}$ value (**Ulfhake and Kellerth, 1984**; **Gustafsson and Pinter, 1984b**). This high value is consistent with the conclusions from **Gustafsson and Pinter, 1984a**, who obtained slightly conservative values for $S_{neuron}$ when approximating the MN surface area as $S_{neuron} = \frac{C}{C_m}$ with $C_m = 1.0 \cdot 10^{-2} F \cdot m^{-2}$.

Finally, the empirical relationship $I_{th} = \frac{2.7 \cdot 10^{-2}}{R^{1.0}}$ (**Table 4**) provides $\Delta V_{th} = 27mV$, consistently with typical reported values (**Brock et al., 1952**; **Eccles et al., 1958a**), despite uncertainties in the value of the membrane resting potential (**Heckman and Enoka, 2012**) and lower values reported by **Gustafsson and Pinter, 1984a**. This supports the conclusions that the relative voltage threshold $\Delta V_{th}$ may be assumed constant within the MN pool in first approximation (**Coombs et al., 1955**; **Gustafsson and Pinter, 1984a**; **Powers and Binder, 2001**), and that Ohm's law is followed in MNs for weak subthreshold excitations (**Glenn and Dement, 1981**; **Ulfhake and Kellerth, 1984**; **Spruston and Johnston, 1992**; **Powers and Binder, 2001**; **Olinder et al., 2006**), as discussed in 'Methods'. However, voltage thresholds $\Delta V_{th}$ tend to be lower in in MNs of large $R$ and long $AHP$ (**Gustafsson and Pinter, 1984a**), suggesting that $\Delta V_{th}$ might be inversely correlated to MN size $S_{MN}$. Moreover, because of the voltage activation of persistent inward currents (PICs) near threshold which add to the external stimulating current and increases the MN input resistance (**Fleshman et al., 1988**), with a prominent effect on small MNs (**Powers and Binder, 2001**), the experimental values of $\Delta V_{th}$ are generally greater than that directly predicted from the product $I_{th} \cdot R$ (**Gustafsson and Pinter, 1984b**), and a variant of Ohm's law such as $\Delta V_{th}(V) = I_{th} \cdot R(V)$ should be considered during MN discharging events. This voltage-dependent variation in the values of $R$ might partly explain why the range of the values of $R$ reported in the literature for weak subthreshold currents (**Appendix 1—table 4**) is generally lower than that of $I_{th}$. While $\Delta V_{th}$ may be always size-dependent, and size-voltage-dependent close to threshold and during firing events, no correlation between $\frac{\Delta V_{th}}{R \cdot I_{th}}$ and $C$, $AHP$, or $ACV$ was found and reported in **Table 4** in subthreshold conditions, consistent with measurements performed by **Gustafsson and Pinter, 1984b** and **Ulfhake and Kellerth, 1984**, substantiating that the dynamics of MN recruitment dominantly rely on $R$, $I^{th}$, and $\Delta V_{th}$ (**Heckman and Enoka, 2012**).

## Relevance of the relationships

*Table 4* provides the first quantitative description of an extensive set of cat experimental data available in the literature. These inter-related relationships were validated, successfully reconciliate the conclusions formulated by the selected experimental studies, and are in strong agreement with the theoretical equations describing Rall's cable theory. This robust framework advances our general understanding of the MN neurophysiology by inferring quantitative relationships for pairs of MN properties that were never concurrently measured in experiments (*Figure 2B*) by crossmatching other relationships involving these properties. For example, $I_{th} = 3.8 \cdot 10^8 \cdot S_{neuron}^{2.5}$ (*Table 4*) quantitatively supports the size dependency of $I_{th}$ that was predicted by **Powers and Binder, 2001** from Rall's equations. Furthermore, modellers can directly and realistically tune with *Table 4* the physiological parameters of phenomenological MN models, such as leaky fire-and-integrate models, and/or build pools of MNs displaying a continuous distribution of realistic MN profiles of MN-specific electrophysiological and morphometric properties, as it has been previously attempted (**Dong et al., 2011**; **Negro et al., 2016**). Models tuned with *Table 4* can, for example, better replicate the discharging behaviour of MNs, obtained from decomposed high-density EMGs, than models scaled with generic property values (**Caillet et al., 2022**). In this view, such tuned models display MN-specific behaviours that should be easier to interpret. The mathematical relationships in *Table 4* can also support future experimental investigations performed on spinal MNs in adult mammals in vivo by completing experimental datasets for which properties that are difficult to measure were not obtained, such as the MN membrane time constant. Experimenters can therefore choose to reduce the workload of measuring

all the MN properties in *Table 1* to favour the identification of a maximum number of MNs and obtain extensive datasets describing larger MN populations that provide an informative window on the continuous distribution of MN properties in an MN pool. The unknown electrophysiological MN properties, typically obtained in vivo, of cadaveric specimens can also be estimated with the size-related relationships in *Table 4* from in vitro measurements of the somal diameter $D_{soma}$ in a slice preparation of the spinal cord. The mathematical relationships in *Table 4* describe the experimental data published in the literature and provide a convenient metric for experimenters to compare their measurements to previous findings.

## Extrapolation of the relationships to other mammals

It is demonstrated in *Figure 7* and with the calculation of the $r^2$ values in *Appendix 1—table 5* that the mathematical equations in *Table 4* obtained from cat data adequately predict the normalized associations between pairs of MN properties in rats and mice. However, the scaling factor $K$ applied to the intercept $k_e$ to scale the normalized relationships $A = K \cdot k_e \cdot B^e$ is species-specific, being, for example, around three times smaller in mice than in cat for the $\{\tau; R\}$ and $\{C; R\}$ pairs (*Figure 7*) and explaining the large *nRMSE* values in these cases. Absolute values of the $\{I_{th}; R\}$ pair were nevertheless accurately predicted in both rats and mice from the cat relationships, as explained by the larger values of $R$ which counterbalances the respectively lower $I_{th}$ values in mice and rats than in cats (*Manuel et al., 2019*; *Highlander et al., 2020*). Despite the age-related data variability (*Highlander et al., 2020*) in the rodent dataset displayed in *Figure 7*, these findings advance our understanding of the systematic inter-mammalian-species variations in MN properties. While the mathematical equations in *Table 4* are specific to cats, the normalized equations in *Appendix 1—table 2* can be scaled with species-specific data if investigating other mammals.

## Henneman's size principle of motor unit recruitment

*Table 5* reports statistically significant power relationships of positive power values between MN and mU indices of size. These results substantiate the concept that $S_{MN}$ and $S_{mU}$ are positively correlated in a motor unit (MU) pool and that large MNs innervate large mUs (*Henneman, 1981*; *Heckman and Enoka, 2012*), a statement that has never been demonstrated from the concurrent direct measurement of $S_{MN}$ and $S_{mU}$. Besides, considering the positive $I_{th} - S_{MN}$ correlation (*Table 4*), and that the mU force recruitment threshold $F^{th}$ is positively correlated to $F^{tet}$ (*Heckman and Enoka, 2012*) and thus to $S_{mU}$ (*Table 2*), larger MUs have both larger current and force recruitment thresholds $I_{th}$ and $F^{th}$ than relatively smaller MUs, which are thus recruited first, consistently with the Henneman's size principle of MU recruitment (*Henneman, 1957*; *Wuerker et al., 1965*; *Henneman et al., 1965a*; *Henneman et al., 1965b*; *Henneman et al., 1974*; *Henneman, 1981*; *Henneman, 1985*). The terminologies 'small MU', 'low-force MU', and 'low-threshold MU' are thus equivalent. Henneman's size principle thus relies on the amplitude of the MN membrane resistance (*Binder et al., 1996*; *Powers and Binder, 2001*; *Heckman and Enoka, 2012*). Finally, the relationships $S_{MN} \propto CV^{1.4} \propto \tau^{-0.7} \propto AHP^{-0.7}$ (*Table 4*) suggest that high-threshold MUs rely on relatively faster MN dynamics, which might partially explain why large MNs can attain relatively larger firing rates than low-thresholds MNs, for example, during ballistic contractions or during events close to maximum voluntary contractions (*Powers and Binder, 2001*; *Heckman and Enoka, 2012*).

It has been repeatedly attempted to extend Henneman's size principle and the correlations between the MU properties in *Table 1* to the concept of 'MU type' (*Burke and ten Bruggencate, 1971*; *Burke, 1981*; *Bakels and Kernell, 1993*; *Powers and Binder, 2001*). While a significant association between 'MU type' and indices of MU size has been observed in some animal (*Fleshman et al., 1981*; *Burke et al., 1982*; *Zengel et al., 1985*) and a few human (*Milner-Brown et al., 1973*; *Stephens and Usherwood, 1977*; *Garnett et al., 1979*; *Andreassen and Arendt-Nielsen, 1987*) studies, it has however not been observed in other animal studies (*Bigland-Ritchie et al., 1998* for a review) and in the majority of human investigations (*Sica and McComas, 1971*; *Goldberg and Derfler, 1977*; *Yemm, 1977*; *Young and Mayer, 1982*; *Thomas et al., 1990*; *Nordstrom and Miles, 1990*; *Elek et al., 1992*; *Macefield et al., 1996*; *Van Cutsem et al., 1997*; *Mateika et al., 1998*; *Fuglevand et al., 1999*; *Keen and Fuglevand, 2004*). Moreover, the reliability of these results is weakened by the strong limitations of the typical MU-type identification protocols. Sag experiments are irrelevant in humans (*Buchthal and Schmalbruch, 1970*; *Thomas et al., 1991*; *Bakels and Kernell, 1993*; *Macefield et al.,*

*1996*; *Bigland-Ritchie et al., 1998*; *Fuglevand et al., 1999*) and lack consistency with other identification methods (*Nordstrom and Miles, 1990*). MU-type identification by twitch contraction time measurements is limited by the strong sources of inaccuracy involved in the transcutaneous stimulation, intramuscular microstimulation, intraneural stimulation, and spike-triggered averaging techniques (*Taylor et al., 2002*; *Keen and Fuglevand, 2004*; *McNulty and Macefield, 2005*; *Negro et al., 2014*; *Dideriksen and Negro, 2018*). Finally, as muscle fibres show a continuous distribution of contractile properties among the MU pool, some MUs fail to be categorized in discrete MU types in some animal studies by histochemical approaches (*Reinking et al., 1975*; *Tötösy de Zepetnek et al., 1992*). Owing to these conflicting results and technical limitations, MU type may not be related to MN size and the basis for MU recruitment during voluntary contractions (*McNulty and Macefield, 2005*; *Duchateau and Enoka, 2011*).

## Limitations

The mathematical relationships derived *Table 4* and the conclusions drawn in 'Discussion' present some limitations, most of which were previously discussed and are summarised here. A first limitation relates to the experimental inaccuracies in measuring the MN properties in *Table 1*. As discussed, the true values of $R$ and $\tau$, and thus of $R_m$ and $C$, may be underestimated because of a membrane leak conductance around the electrode and some voltage-activated membrane nonlinearities. Yet, the selected studies reduced the impact of these phenomena, which besides may not affect the association between MN properties (*Gustafsson and Pinter, 1984a*). Also, $\tau$, $C$, and $R_m$ were estimated assuming that the MN membrane is isopotential. This is valid in the soma, where $\tau$, $C$, and $R_m$ were measured, and the relationships in *Table 4* are adequate for an MN simplified to one equivalent cable. However, as the membrane resistivity may increase with somatofugal distance, as discussed previously, the relationships in *Table 4* may be offset towards larger $R$ values in dendritic areas, which should be accounted for when modelling the MN dendrites in compartmental MN models or completely reconstructed dendritic trees. Finally, the selected cat studies are relatively dated, as observed in *Manuel et al., 2019*. Thus, recent computer-assisted techniques of MN morphometric measurements were notably not applicable in those studies, yielding important sources of inaccuracies, as discussed in Appendix 1. Therefore, the relationships involving $S_{neuron}$ are subjected to a higher level of inaccuracy. A second limitation arises from the inter-study variability of the experimental datasets reported in the selected studies. As discussed, animal preparations slightly diverged between the selected studies. Also, because of the scarcity of available data in the literature, measurements obtained from MNs innervating different muscles from adult animals of different sex and age were merged into unique datasets. This inter-study variability, added to the 'unexplained' variance reported in *Highlander et al., 2020*, was however reduced by the normalization of the experimental datasets, as investigated in *Appendix 1—figures 2 and 4*. The normalization approach is however only valid if the experimental studies identified the same largest MN relatively to the MN populations under investigation. This cannot be verified but is likely as most studies returned similar normalized distributions of their measured parameters, as displayed in *Appendix 1—figure 1*, inferring that a similar portion of the MN pools was identified in the selected studies. Finally, the results proposed in this study may be affected by methodological bias from the three research groups that provided most of the data points processed in this study, even if all the selected studies reported similar methods to measure the properties, as discussed in 'Methods'.

A third limitation is due to the limited processable experimental data available in the literature, especially in rats and mice, as also discussed by *Heckman and Enoka, 2012*. The extrapolation of the relationships in *Table 4* to rat and mouse species could only be assessed against a small set of studies and for three associations only. The conclusions on the associations between $S_{mU}$ and $S_{MN}$ also relied on few studies providing few measurements due to the considerable amount of work required to measure both MN and mU properties in single MUs. In cats, some pairs of MN properties were investigated in only one study, preventing inter-study comparisons, and stressing the usefulness of *Table 4* for reconciliating the data available in the literature. Within the experimental datasets, the scarcity of data points in some regions of the data distributions (e.g., see the skewed $I_{th}$ and $R$ distributions in the density histograms in *Appendix 1—figure 1*) transfers during processing to the final size-dependent datasets (*Figure 4*). This data heterogeneity may affect the reliability of the relationships in *Table 4* for the extreme MN sizes (see, in *Figure 4*, the 95% confidence interval of the regressions widening for

small $S_{MN}$ in the $\{AHP; S_{MN}\}$ and $\{R; S_{MN}\}$ datasets and for large $S_{MN}$ in the $\{I_{th}; S_{MN}\}$ dataset). In the validation procedure, the highest $nME$ values reported in *Figure 6* were obtained for these regions of limited data for most of the global datasets. This heterogeneous density of the data may be explained by the natural skewness of the MN property distributions in the MN pool towards many small MNs and few large MNs (*Heckman and Enoka, 2012*) and/or a systematic experimental bias towards identifying and investigating relatively large MUs.

A fourth limitation arises from the relatively low $r^2$ values obtained from the power regressions in *Table 3*. Although these power trendlines are statistically significant (see p-values in *Table 3*) and globally provide a better description of the data than other fits (see *Appendix 1—table 1*), they cannot entirely describe the associations between MN property distributions, in line with the conclusions reported in the selected studies. According to these first four limitations, the mathematical relationships in *Table 4* best reproduce published cat data but include the level of inaccuracies of the original experimental approaches.

A fifth limitation is the restriction of the study to the MN properties reported in *Table 1*. Other MN-specific properties, such as the electronic length, the relationship between discharge rate and excitatory current and its hysteresis, or the amplitude of the inhibitory and excitatory postsynaptic potentials, were omitted because they were rarely measured or reported along with $S_{MN}$, $ACV$, $AHP$, $R$, $I_{th}$, $C$, and $\tau$ in the selected studies. Moreover, as the cell geometry was not investigated in this study, other MN properties (*Clements and Redman, 1989*) such as the effects of the distributed synaptic integration along the dendritic tree were overlooked. Finally, apart from $I_{th}$, $ACV$, and $AHP$, all the MN properties investigated in this study are passive properties, that is, baseline properties of the cell at rest (*Powers and Binder, 2001*), as they were measured using weak sub-threshold current pulses. Therefore, other MN active properties, such as voltage-dependent ionic channel-related properties including PICs, which activate close or above threshold, were not considered in this study. While this limits the applicability of *Table 4* to a functional context or in comprehensive Hodgkin–Huxley-like models of MNs, some relationships in *Table 4* remain pertinent in those conditions if they are linked with other observations in the literature on MN active properties, for example, MNs of small $S_{MN}$ having longer lasting total PICs of greater hyperpolarized activation than MNs of large $S_{MN}$ (*Heckman and Enoka, 2012*). Despite the aforementioned limitations, the relationships in *Table 4* can be directly used in phenomenological RC approaches like LIF models (*Izhikevich, 2004*; *Teeter et al., 2018*), which rely on the properties reported in *Table 1*, to derive profiles of inter-consistent MN-specific properties and describe realistic continuous distributions of the MN properties $R$, $I_{th}$, $C$, $\Delta V_{th}$, and $\tau$ in the MN pool, as attempted in previous works (*Dong et al., 2011*; *Negro et al., 2016*). An example of a computational modelling application of *Table 4* is provided in *Caillet et al., 2022*, where the relationships tune a cohort of LIF models which estimate, from the firing activity of a portion of the MN pool obtained from decomposed high-density EMGs, a realistic distribution of the MN properties in the MN pool and the firing behaviour of a complete MN pool in an isometrically contracting human muscle.

As a final limitation, the relationships in *Table 4* were obtained from a regression analysis and therefore provide correlations between some MN properties in an MN pool but cannot be used to draw conclusions on the causality behind these associations.

## Conclusion

This study provides in *Table 4* a mathematical framework of quantitative associations between the MN properties $S_{MN}, ACV$, $AHP$, $R$, $R_m$, $I_{th}$, $C$, and $\tau$. This framework, which is consistent with most of the empirical and theoretical conclusions in the literature, clarifies our understanding of the association between these MN properties and constitutes a convenient tool for neuroscientists, experimenters, and modellers to generate hypotheses for experimental studies aiming at investigating currently unreported relationships, support experimentations, and build virtual MN profiles of inter-consistent MN-specific properties for MN modelling purposes.

## Acknowledgements

AC was supported by the Skempton Scholarship and LM by an Imperial College Research Fellowship granted by Imperial College London. The authors want to thank Dr Marin Manuel for making available

the experimental data published in Huh et al., 2021, Time course of alterations in adult spinal moto-neuron properties in the SOD1 (G93A) mouse model of ALS, Eneuro.

## Additional information

### Funding

| Funder | Grant reference number | Author |
|---|---|---|
| Imperial College London | Skempton Scholarship | Arnault H Caillet |
| Imperial College London | Imperial College Research Fellowship | Luca Modenese |

The funders had no role in study design, data collection and interpretation, or the decision to submit the work for publication.

### Author contributions

Arnault H Caillet, Conceptualization, Data curation, Formal analysis, Validation, Methodology, Writing – original draft, Writing – review and editing; Andrew TM Phillips, Supervision, Writing – review and editing; Dario Farina, Conceptualization, Supervision, Writing – review and editing; Luca Modenese, Supervision, Methodology, Writing – review and editing

### Author ORCIDs

Arnault H Caillet  http://orcid.org/0000-0001-6146-1829
Andrew TM Phillips  http://orcid.org/0000-0001-6618-0145
Dario Farina  http://orcid.org/0000-0002-7883-2697
Luca Modenese  http://orcid.org/0000-0003-1402-5359

### Decision letter and Author response

Decision letter https://doi.org/10.7554/eLife.76489.sa1
Author response https://doi.org/10.7554/eLife.76489.sa2

## Additional files

### Supplementary files
• Transparent reporting form

### Data availability

Figure 3 - Source Data 1 contains the numerical data used to generate Figure 3. Figure 7 - Source Data 1 and Figure 7 - Source Data 2 contain the numerical data used to generate Figure 7. Table 5 - Source Data 1 contains the numerical data used to compute the mathematical relationships presented in Table 5. Please note that our study used exclusively data from previous investigations, for which public datasets were not available. The data were manually digitised by the authors from published figures and are made available as supplementary materials.

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

## Appendix 1

### Inter-study data variability

Inter-study variability greater than the threshold defined in 'Methods' was observed in three, two, and four property distributions for the metrics range, mean, and CoV metrics, respectively. Indeed, sensibly different ranges of property values were observed to be reported between studies of the $\{ACV; S_{MN}\}$, $\{AHP; ACV\}$, and $\{R; AHP\}$ datasets for the $S_{MN}$, $ACV$, and $AHP$ properties, respectively, with $sd_g \in [10; 15]$ % and/or $\frac{sd_g}{mean_g} \in [0.15; 0.33]$. Also, the studies of the $\{R; AHP\}$ and $\{\tau; R\}$ datasets returned some variability in the mean of the reported data for properties $AHP$ and $R$, respectively, with $sd_g \in [10; 11]$ % and/or $\frac{sd_g}{mean_g} \in [0.15; 0.23]$. The $\{ACV; S_{MN}\}$, $\{R; ACV\}$, and $\{R; AHP\}$ global datasets showed variable data dispersion around the mean for properties $S_{MN}$, $R$ and $ACV$, and $AHP$, respectively, with $sd_g \in [10; 11]$ % and/or $\frac{sd_g}{mean_g} \in [0.15; 0.30]$.

### Discussion on the potential underestimation of $q_S^E$

The minimum and maximum values of $S_{neuron}$ and $D_{soma}$ reported in *Appendix 1—table 1* may underestimate the true distribution of MN sizes in an MN pool for several reasons, and thus underestimate the true value of $q_S^E$. First, the studies reported in *Appendix 1—table 1* investigated MN populations of small size, so that the largest and smallest MNs of the investigated MN pools may not have been systematically identified and measured. Also, dendritic labelling by staining may only identify dendrites up to the third dendritic branch (*Issa et al., 2010*; *Brandenburg et al., 2018*) and full dendritic trees may not be identified, thus underestimating $S_{neuron}$ for the largest MNs that exhibit the highest dendritic tree complexity (*Brandenburg et al., 2020*). Finally, the distribution of gamma- and alpha-MN sizes is typically bimodal and the size distributions of the two populations are reported to overlap (*Moschovakis et al., 1991*; *Vult von Steyern et al., 1999*; *Friese et al., 2009*; *Deardorff et al., 2013*). However, most studies reported in *Appendix 1—table 1* identify the alpha MNs from the gamma MNs by visually splitting the bimodal distributions of MN sizes in two independent domains (*Donselaar et al., 1986*; *Ishihara et al., 2001*), thus neglecting the overlap. Similarly, some studies identify as gamma MNs any MN showing an axonal conduction velocity typically less than $\sim 60 m \cdot s^{-1}$ (*Zwaagstra and Kernell, 1980*; *Ulfhake and Kellerth, 1984*). These approaches lead to overestimating the lower limit for the reported $D_{soma}$ and $S_{neuron}$ values. Consequently, true ranges in MN sizes may be larger than the theoretical ones reported in *Appendix 1—table 1*. A similar comment can be made for the electrophysiological properties reported in *Appendix 1—table 2*. However, the $q_S^E = 2.4$-fold range obtained for cats is consistent with the fold ranges obtained from rat and mouse data for $D_{soma}$, as reported in *Appendix 1—table 1*, while it underestimates the 4.4-fold range in $S_{neuron}$ values observed in mice. *Appendix 1—table 1* finally shows that typical cat MN sizes are larger than in rats and mice, as reviewed by *Manuel et al., 2019*.

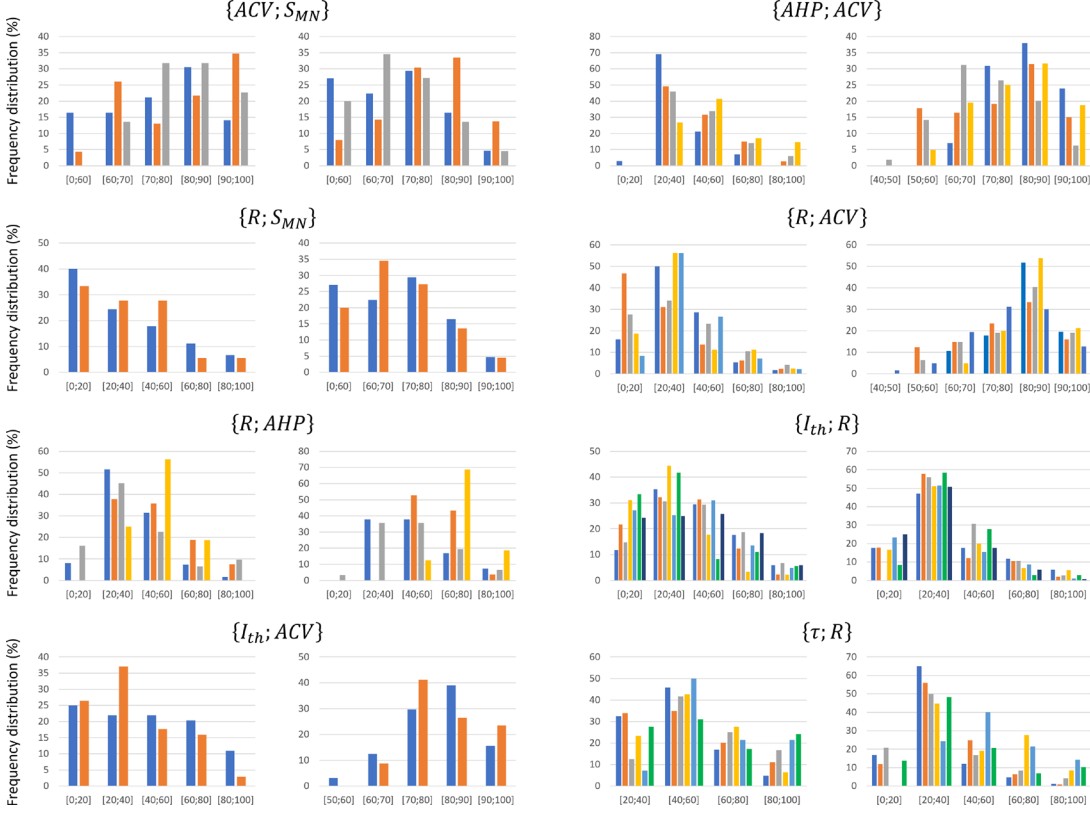

**Appendix 1—figure 1.** Density histograms of the data distributions reported in the experimental studies included in the global datasets. Depending on the typical range over which each property spans, the distributions are divided in steps of 10 or 20%. The frequency distribution is provided in percentage of the total number of reported data points in a study. Different studies are displayed with different colours in each graph. For each dataset $\{A; B\}$, the frequency distributions of property $A$ and $B$ are provided in the left and right plot, respectively.

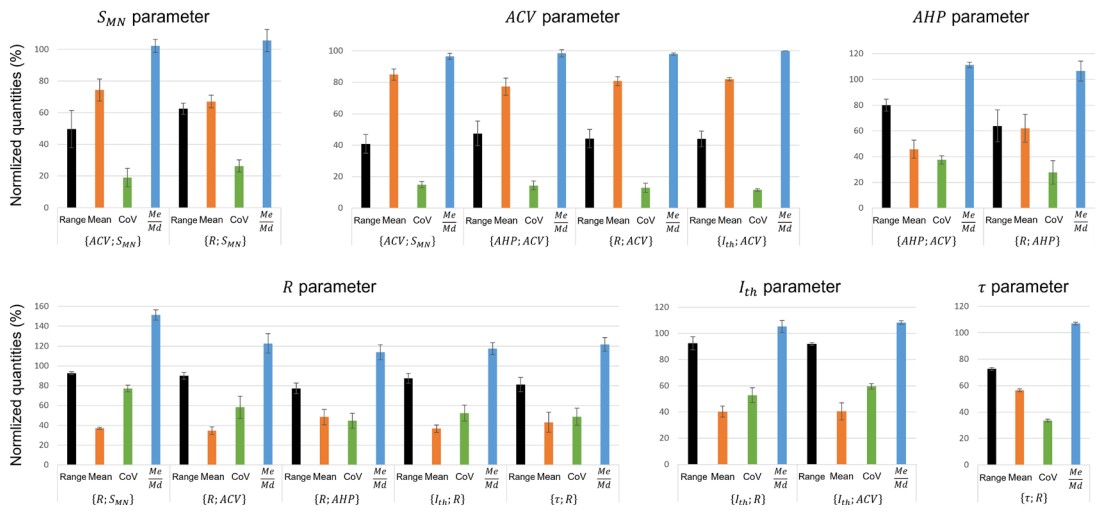

**Appendix 1—figure 2.** Assessment of data variability between the experimental studies that constitute the eight global datasets that include at least two experimental studies. For each experimental study included in the global dataset $\{A; B\}$, the range, mean, coefficient of variation ($CoV = \frac{standard\ deviation}{Mean}$), and the ratio $\frac{Me}{Md} = \frac{Mean}{Median}$ of the experimental $A$ values measured in this study were computed. Then, the average (bars) and standard deviation (error bars) across experimental studies of these four metrics (range, mean, CoV, $\frac{Me}{Md}$) were calculated for each global dataset independently. For example, the first bar in the '$S_{MN}$ parameter' plot represents the average across three studies (**Kernell and Zwaagstra, 1981**; **Cullheim, 1978**; **Burke et al., 1982**) of the range of $S_{MN}$

*Appendix 1—figure 2 continued on next page*

*Appendix 1—figure 2 continued*

values reported in these studies, for the global dataset $\{ACV; S_{MN}\}$. The computed standard deviations (error bars) express for each global dataset the inter-study variability of (1) the length of the identified bandwidth of the motoneuron (MN) pool (range metric), (2) the spread of values around the mean (CoV metric), (3) the skewness of the distributions, and (4) whether the distributions from different studies are centred (mean metric). A global dataset $\{A; B\}$ reporting narrow error bars for parameter $A$ for the four metrics infers that the experimental studies constituting this dataset measured similar distributions of property $A$.

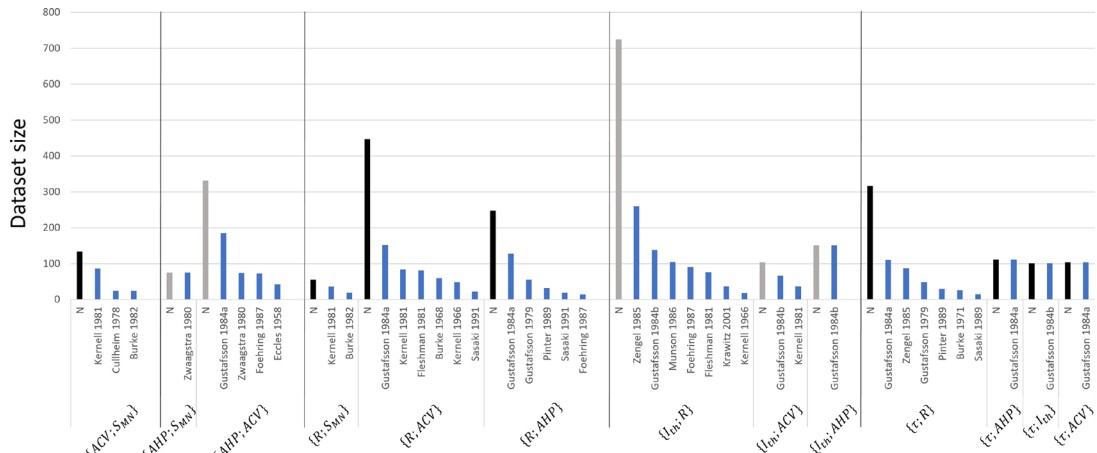

**Appendix 1—figure 3.** Distribution of the size of the experimental datasets constituting the global datasets for assessment of the variability in the input data. The histogram is divided between global datasets (half vertical lines), grouped as final size-dependent datasets (full vertical lines). For each global dataset, the total number of data points is reported (label: 'N'), while the size of the constitutive experimental studies is reported in decreasing order.

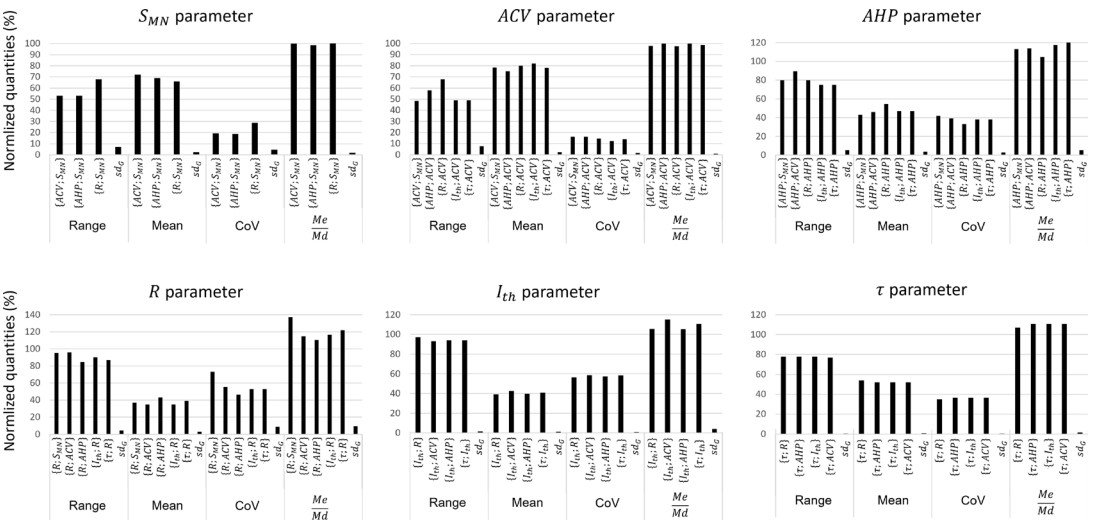

**Appendix 1—figure 4.** Assessment of data variability between the global datasets. This figure compares the distributions of the $S_{MN}$, $ACV$, $AHP$, $R$, $I_{th}$, and $\tau$ properties between the normalized global datasets built in this study. For each property, the range, mean, coefficient of variation ($oV = \frac{standard\ deviation}{Mean}$), and the ratio $\frac{Me}{Md} = \frac{Mean}{Median}$ of its distribution in a global dataset is compared to the other global datasets it appears in. For each property, the standard deviation $sd_G$ between global datasets of these four metrics is computed. A low $sd_G$ value reflects the low variability of the property distribution between global datasets.

**Appendix 1—table 1.** $r^2$ values obtained for each experimental dataset $\{A; B\}$ when performing a linear regression analysis on $\{A; B\}$ directly ('Linear') and on the $ln(A) - ln(B)$ ('Power') and $ln(A) - B$ ('Exponential') transformations of $\{A; B\}$.

The $r^2$ values returned by the three types of regression cannot be directly compared to estimate the best model. However, the power fit returned $r^2 > 0.5$ for relatively more experimental datasets than the linear and exponential fits.

| | Studies | Linear | Power | Exponential |
|---|---|---|---|---|
| | *Cullheim, 1978* | 0.42 | 0.45 | 0.43 |
| | *Kernell and Zwaagstra, 1981* | 0.62 | 0.63 | 0.6 |
| $\{ACV; S_{MN}\}$ | *Burke et al., 1982* | 0.23 | 0.26 | 0.24 |
| $\{AHP; S_{MN}\}$ | *Zwaagstra and Kernell, 1980* | 0.33 | 0.37 | 0.35 |
| | *Eccles et al., 1958b* | 0.53 | 0.54 | 0.54 |
| | *Zwaagstra and Kernell, 1980* | 0.42 | 0.44 | 0.42 |
| | *Gustafsson and Pinter, 1984a* | 0.71 | 0.68 | 0.7 |
| $\{AHP; ACV\}$ | *Foehring et al., 1987* | 0.15 | 0.14 | 0.14 |
| | *Kernell and Zwaagstra, 1981* | 0.68 | 0.71 | 0.7 |
| $\{R; S_{MN}\}$ | *Burke et al., 1982* | 0.41 | 0.5 | 0.46 |
| | *Kernell, 1966* | 0.70 | 0.65 | 0.64 |
| | *Burke, 1968* | 0.28 | 0.31 | 0.31 |
| | *Kernell and Zwaagstra, 1981* | 0.68 | 0.71 | 0.71 |
| | *Fleshman et al., 1981* | 0.3 | 0.21 | 0.2 |
| | *Gustafsson and Pinter, 1984a* | 0.45 | 0.44 | 0.42 |
| $\{R; ACV\}$ | *Sasaki, 1991* | 0.68 | 0.68 | 0.66 |
| | *Gustafsson, 1979* | 0.43 | 0.41 | 0.42 |
| | *Gustafsson and Pinter, 1984a* | 0.71 | 0.68 | 0.67 |
| | *Foehring et al., 1987* | 0.33 | 0.32 | 0.33 |
| | *Pinter and Vanden Noven, 1989* | 0.66 | 0.53 | 0.59 |
| $\{R; AHP\}$ | *Sasaki, 1991* | 0.57 | 0.49 | 0.52 |
| | *Kernell, 1966* | 0.52 | 0.52 | 0.68 |
| | *Fleshman et al., 1981* | 0.28 | 0.34 | 0.38 |
| | *Gustafsson and Pinter, 1984b* | 0.63 | 0.83 | 0.83 |
| | *Zengel et al., 1985* | 0.49 | 0.6 | 0.62 |
| | *Munson et al., 1986* | 0.34 | 0.42 | 0.47 |
| | *Foehring et al., 1987* | 0.34 | 0.47 | 0.47 |
| $\{I_{th}; R\}$ | *Krawitz et al., 2001* | 0.32 | 0.49 | 0.44 |
| | *Kernell and Zwaagstra, 1981* | 0.23 | 0.24 | 0.24 |
| $\{I_{th}; ACV\}$ | *Gustafsson and Pinter, 1984b* | 0.4 | 0.52 | 0.5 |

*Appendix 1—table 1 Continued on next page*

*Appendix 1—table 1 Continued*

| | Studies | Linear | Power | Exponential |
|---|---|---|---|---|
| $\{I_{th}; AHP\}$ | *Gustafsson and Pinter, 1984b* | 0.55 | 0.72 | 0.69 |
| $\{C; R\}$ | *Gustafsson and Pinter, 1984a* | 0.55 | 0.57 | 0.57 |
| $\{C; I_{th}\}$ | *Gustafsson and Pinter, 1984a* | 0.23 | 0.34 | 0.24 |
| $\{C; AHP\}$ | *Gustafsson and Pinter, 1984a* | 0.25 | 0.26 | 0.26 |
| $\{C; ACV\}$ | *Gustafsson and Pinter, 1984a* | 0.17 | 0.22 | 0.19 |
| | *Burke and ten Bruggencate, 1971* | 0.04 | 0.07 | 0.03 |
| | *Gustafsson, 1979* | 0.6 | 0.63 | 0.61 |
| | *Gustafsson and Pinter, 1984a* | 0.63 | 0.69 | 0.59 |
| | *Zengel et al., 1985* | 0.39 | 0.33 | 0.36 |
| | *Pinter and Vanden Noven, 1989* | 0.65 | 0.69 | 0.62 |
| $\{\tau; R\}$ | *Sasaki, 1991* | 0.68 | 0.72 | 0.64 |
| $\{\tau; I_{th}\}$ | *Gustafsson and Pinter, 1984b* | 0.63 | 0.59 | 0.57 |
| $\{\tau; ACV\}$ | *Gustafsson and Pinter, 1984a* | 0.69 | 0.76 | 0.76 |
| $\{\tau; AHP\}$ | *Gustafsson and Pinter, 1984a* | 0.34 | 0.32 | 0.31 |

**Appendix 1—table 2.** Mathematical empirical normalized relationships between the motoneuron (MN) properties $D_{soma}$, $R$, $C$, $\tau$, $I_{th}$, $AHP$, and $ACV$. Each column provides the relationships between one and the six other MN properties. All constants and properties are normalized up to a theoretical 100% maximum value. To scale the normalized relationships $A = k \cdot B^c$ for a specific mammalian species, the normalized intercept $k$ must be scaled with a pair of values for properties $A$ and $B$ obtained from the same motoneuron in that species.

| | $D_{soma}$ | $R$ | $C$ | $\tau$ | $I_{th}$ | $AHP$ | $ACV$ |
|---|---|---|---|---|---|---|---|
| $D_{soma}$ | | $R = \frac{9.6 \cdot 10^5}{D_{soma}^{2.4}}$ | $C = 1.2 \cdot D_{soma}$ | $\tau = \frac{2.6 \cdot 10^4}{D_{soma}^{1.5}}$ | $I_{th} = 9.0 \cdot 10^{-4} \cdot D_{soma}^{2.5}$ | $AHP = \frac{2.5 \cdot 10^4}{D_{soma}^{1.5}}$ | $ACV = 4.0 \cdot D_{soma}^{0.7}$ |
| $R$ | $D_{soma} = \frac{2.9 \cdot 10^2}{R^{0.4}}$ | | $C = \frac{2.7 \cdot 10^2}{R^{0.4}}$ | $\tau = 5.8 \cdot R^{0.6}$ | $I_{th} = \frac{1.5 \cdot 10^3}{R}$ | $AHP = 4.7 \cdot R^{0.6}$ | $ACV = \frac{2.0 \cdot 10^2}{R^{0.3}}$ |
| $C$ | $D_{soma} = 8.6 \cdot 10^{-1} \cdot C$ | $R = \frac{1.4 \cdot 10^6}{C^{2.5}}$ | | $\tau = \frac{3.3 \cdot 10^4}{C^{1.5}}$ | $I_{th} = 6.2 \cdot 10^{-4} \cdot C^{2.6}$ | $AHP = \frac{3.1 \cdot 10^4}{C^{1.6}}$ | $ACV = 3.6 \cdot C^{0.7}$ |
| $\tau$ | $D_{soma} = \frac{9.5 \cdot 10^2}{\tau^{0.7}}$ | $R = 5.6 \cdot 10^{-2} \cdot \tau^{1.6}$ | $C = \frac{8.5 \cdot 10^2}{\tau^{0.7}}$ | | $I_{th} = \frac{2.9 \cdot 10^4}{\tau^{1.7}}$ | $AHP = 7.8 \cdot 10^{-1} \cdot \tau$ | $ACV = \frac{4.6 \cdot 10^2}{\tau^{0.5}}$ |
| $I_{th}$ | $D_{soma} = 1.6 \cdot 10^1 \cdot I_{th}^{0.4}$ | $R = \frac{1.1 \cdot 10^3}{I_{th}}$ | $C = 1.7 \cdot 10^1 \cdot I_{th}^{0.4}$ | $\tau = \frac{4.2 \cdot 10^2}{I_{th}^{0.6}}$ | | $AHP = \frac{3.7 \cdot 10^2}{I_{th}^{0.6}}$ | $ACV = 2.7 \cdot 10^1 \cdot I_{th}^{0.3}$ |
| $AHP$ | $D_{soma} = \frac{8.1 \cdot 10^2}{AHP^{0.7}}$ | $R = 8.4 \cdot 10^{-2} \cdot AHP^{1.6}$ | $C = \frac{7.3 * 10^2}{AHP^{0.6}}$ | $\tau = 1.3 \cdot AHP$ | $I_{th} = \frac{1.9 * 10^4}{AHP^{1.7}}$ | | $ACV = \frac{4.1 \cdot 10^2}{AHP^{0.5}}$ |
| $ACV$ | $D_{soma} = 1.4 \cdot 10^{-1} \cdot ACV^{1.4}$ | $R = \frac{1.2 \cdot 10^8}{ACV^{3.5}}$ | $C = 1.7 \cdot 10^{-1} \cdot ACV^{1.4}$ | $\tau = \frac{5.0 \cdot 10^5}{ACV^{2.1}}$ | $I_{th} = 5.9 \cdot 10^{-6} \cdot ACV^{3.6}$ | $AHP = \frac{5.0 \cdot 10^5}{ACV^{2.2}}$ | |

**Appendix 1—table 3.** Typical ranges of physiological values for $D_{soma}$ and $S_{neuron}$ in cat rat and mouse species.

$S_{MN}$ is found to vary over an average $q_S^E = 2.4$-fold range, which sets the amplitude of the theoretical ranges. Absolute {min; max} reports the minimum and maximum values retrieved in the reference studies for $D_{soma}$ and $S_{neuron}$, while average {min; max} is obtained as the average across reference studies of minimum and maximum values retrieved per study.

| | Property | Unit | Absolute {min;max} | Average {min; max} | $q_S^E$ | Reference studies | Theoretical range |
|---|---|---|---|---|---|---|---|
| Cat | $D_{soma}$ | [μm] | {25.0; 90.0} | {36.7; 75.5} | 2.2 | Kernell, 1966; Cullheim, 1978; Zwaagstra and Kernell, 1980; Kernell and Zwaagstra, 1981; Ulfhake and Kellerth, 1981; Zwaagstra and Kernell, 1981; Burke et al., 1982; Donselaar et al., 1986; Destombes et al., 1992 | [33; 79] |
| | $S_{neuron}$ | [mm²] | {0.08; 0.64} | {0.17; 0.45} | 2.7 | Barrett and Crill, 1974; Ulfhake and Kellerth, 1981; Burke et al., 1982; Ulfhake and Kellerth, 1984; Ulfhake and Cullheim, 1988; Moschovakis et al., 1991 | [0.18; 0.44] |
| Rat | $D_{soma}$ | [μm] | {17.5; 66.0} | {23.9; 55.7} | 2.4 | Swett et al., 1986; Vult von Steyern et al., 1999; Copray and Kernell, 2000; Ishihara et al., 2001; Deardorff et al., 2013; Mierzejewska-Krzyżowska et al., 2014 | |
| Mouse | $D_{soma}$ | [μm] | {14.0; 35.0} | {14.0; 35.0} | 2.5 | Vult von Steyern et al., 1999 | |
| | $S_{neuron}$ | [mm²] | {0.01; 0.08} | {0.01; 0.06} | 4.4 | Amendola and Durand, 2008; Brandenburg et al., 2020 | |

**Appendix 1—table 4.** Typical ranges of physiological values for the motoneuron (MN) properties $R$, $R_m$, $C$, $\tau$, $I_{th}$, $AHP$, and $ACV$.

As described in 'Methods', $q_A^E$ is the average among reference studies of the ratios of minimum and maximum values; the properties experimentally vary over a $q_A^E$-fold range. This ratio compares with the theoretical ratio $q_A^T = \left(q_S^E\right)^{|c|}$ (with $c$ taken from *Table 3*), which sets the amplitude of the theoretical ranges. Absolute and average {min; max} are obtained as described in the main sections.

| MN property | Unit | Absolute exp {min;max} | Average exp {min; max} | $q_A^E$ | Reference studies | $q_A^T$ | $\frac{q_A^E}{q_A^T}$ | Theoretical range |
|---|---|---|---|---|---|---|---|---|
| $R$ | [MΩ] | {0.2; 8.1} | {0.4; 4.0} | 10.9 | Kernell, 1966; Burke, 1968; Burke and ten Bruggencate, 1971; Barrett and Crill, 1974; Gustafsson, 1979; Kernell and Zwaagstra, 1981; Fleshman et al., 1981; Burke et al., 1982; Ulfhake and Kellerth, 1984; Gustafsson and Pinter, 1984a; Zengel et al., 1985; Foehring et al., 1986; Munson et al., 1986; Foehring et al., 1987; Sasaki, 1991; Krawitz et al., 2001 | 8.4 | 1.3 | [0.5; 4.0] |
| $C$ | [nF] | {2.2; 8.5} | {2.2; 8.5} | 3.9 | Gustafsson and Pinter, 1984a; Gustafsson and Pinter, 1985 | 2.3 | 1.6 | [3.2; 7.5] |
| $\tau$ | [ms] | {2.0; 14.2} | {2.9; 10.2} | 3.5 | Burke and ten Bruggencate, 1971; Barrett and Crill, 1974; Gustafsson, 1979; Ulfhake and Kellerth, 1984; Gustafsson and Pinter, 1984a; Gustafsson and Pinter, 1985; Sasaki, 1991 | 3.7 | 0.9 | [2.8; 10.3] |
| $I_{th}$ | [nA] | {1.7; 52.7} | {2.3; 36.6} | 16.3 | Fleshman et al., 1981; Kernell and Monster, 1981; Ulfhake and Kellerth, 1984; Gustafsson and Pinter, 1984b; Zengel et al., 1985; Foehring et al., 1986; Munson et al., 1986; Foehring et al., 1987; Krawitz et al., 2001 | 9.1 | 1.8 | [3.9; 35.0] |
| $AHP$ | [ms] | {10.6; 266.6} | {44.2; 158.7} | 4.1 | Eccles et al., 1957; Gustafsson, 1979; Dum and Kennedy, 1980; Zwaagstra and Kernell, 1980; Ulfhake and Kellerth, 1984; Gustafsson and Pinter, 1984a; Zengel et al., 1985; Foehring et al., 1987; Sasaki, 1991 | 3.8 | 1.1 | [42.6; 160.2] |
| $ACV$ | [m·s-1] | {51.1; 127.2} | {65.3; 114.8} | 1.8 | Eccles et al., 1957; Mcphedran et al., 1965; Kernell, 1966; Appelberg and Emonet-Dénand, 1967; Burke, 1968; Barrett and Crill, 1974; Proske and Waite, 1974; Bagust, 1974; Stephens and Stuart, 1975; Cullheim, 1978; Dum and Kennedy, 1980; Zwaagstra and Kernell, 1980; Kernell and Zwaagstra, 1981; Fleshman et al., 1981; Glenn and Dement, 1981; Burke et al., 1982; Gustafsson and Pinter, 1984a; Zengel et al., 1985; Foehring et al., 1986; Foehring et al., 1987 | 2.3 | 0.8 | [63.5; 116.6] |

**Appendix 1—table 5.** Validation of the cat relationships (Table 4) against rat and mouse data. nME, normalized maximum error; nRMSE, normalized root mean square error; $r^2_{pred}$, coefficient of determination between experimental and predicted quantities; $r^2_{exp}$, coefficient of determination of the power trendline directly fitted to the experimental data.

| Animal | Dataset | Reference studies | *nME* (%) | *nRMSE* (%) | $r^2_{pred}$ | $r^2_{exp}$ |
|---|---|---|---|---|---|---|
| Rat | $\{I_{th};R\}$ | *Gardiner, 1993*; *Bakels and Kernell, 1993*; *Lee and Heckman, 1998*; *Button et al., 2008*; *Turkin et al., 2010*; *Krutki et al., 2015* | 352 | 16 | 0.53 | 0.54 |
| | $\{I_{th};R\}$ | *Delestrée et al., 2014*; *Martínez-Silva et al., 2018*; *Huh et al., 2021* | 385 | 16 | 0.46 | 0.46 |
| | $\{\tau;R\}$ | *Manuel et al., 2009* | 1219 | 168 | 0.35 | 0.40 |
| Mouse | $\{C;R\}$ | *Manuel et al., 2009* | 1915 | 98 | 0.38 | 0.38 |

