## [Editor Report]

This article will be of interest to scientists studying the contribution of motoneuron behaviour to motor control as well as anyone interested in the relation between neuron morphology, intrinsic properties, and neuron behaviour. The authors have distilled decades of research on motoneuron properties into a set of mathematical relationships that can guide both experimentalists and modellers interested in developing realistic models of populations of motoneurons. In fact, Caillet et al. present a data-driven regression analysis to infer the relationships between morphological and electrophysiological measures from spinal motor neurons in different animal species. Finally, the authors emphasize the value of this approach, but also carefully consider its limitations, including inter-study variability and limited sample sizes in the experimental datasets used to derive the relationships between multiple intrinsic properties.

---

## [Decision Letter]

**Decision letter after peer review:**

Thank you for submitting your article "Mathematical Relationships between Spinal Motoneuron Properties" for consideration by *eLife*. Your article has been reviewed by 3 peer reviewers, one of whom is a member of our Board of Reviewing Editors, and the evaluation has been overseen by Barbara Shinn-Cunningham as the Senior Editor. The following individuals involved in review of your submission have agreed to reveal their identity: Randall K Powers (Reviewer #2); Leonardo Abdala Elias (Reviewer #3).

Essential revisions:

1. The format of the paper should follow the structure of *eLife*. The legend of Figure 1 should describe very explicitly the steps used to create the final size-related datasets, maybe giving an example. The circles and symbols in Figure 2 should be larger.

2. The Introduction should be more inclusive and respectful with previous studies. It is unfair to state that some inferences are "speculative" since some authors provide linear regression curves (Eccles et al., 1958, for instance) while others consider the theoretical analysis by Rall. Also, when mentioning computational models, since the present study is focused on motor neurons, consider citing the extensive literature on models of motor neurons (see p. 3, l. 51 and 52).

3. In both Intro and Discussion, the authors should be careful when declaring the relevance of the current study. Several statements cannot be supported by analysis, for example, the comments that the method "can accelerate future research in the behaviour of individual MNs" and that mathematical equations can be scaled "to estimate the MN property values that cannot be measured in humans in vivo". How can you scale the equations without prior information on human motor neuron electrophysiology and morphology?

4. The methods section could be expanded to explain some of the measured electrophysiological properties in more detail. Most of the direct measurements (ACV, AHP duration, RN, Ith) are fairly straightforward, with the possible exception of time constant (see comments in Public Review). However, the derived measurements need to be explained in the Methods section. Both estimates of specific membrane resistivity (Rm) and whole cell capacitance (C) can be estimated using an equivalent cable model of the motoneuron along with an estimate of electrotonic length, and the formulas for these should be stated (see Ulfhake and Kellerth, Brain Res, 1984 equations 2 and 3 and Gustafsson and Pinter, JPhysiol, 1984a, equation 1). It should be mentioned in Methods that whole cell capacitance can be used as an estimate of cell surface area by assuming a value for specific capacitance of 1 microF/cm2 and that these area estimates are roughly in line with direct measurements although they tend to be a bit high (see Figure 6 of Gustafsson and Pinter 1984a).

5. Data used in the analysis are both from in vivo and in vitro recordings, not exclusively from in vivo experiments as declared in several parts of the manuscript (e.g., Abstract). Morphometric data cannot be recorded in vivo.

6. The authors should revise their method to include some information on data variability. A simple way is to provide the confidence interval of the regression. In this case, all parameters of the mathematical equations will have a range (95% confidence interval). The values provided in the current manuscript are only average values.

7. It is not clear why the authors used a 70-30% scheme for crossvalidation and why only 5 validations were performed. A larger set of validations are probably needed and at least another training-testing proportion should be carried out.

8. Tables 4, 5 and 7 could be moved to the Supplementary material section.

9. In the merged data sets shown in Figure 4 it would be useful to use different symbols for each data set (i.e., different symbols for {AHP;Smn}1 and {AHP;Smn}2).

10. There is no oval box with fc(C) in Figure 1. Please, revise. Also, consider changing Figure 1 with a pseudo-code of the algorithm to improve clarity.

11. Ith was estimated using different approaches in the selected studies. Some studies have used triangular-shaped currents, while others used step currents. How would different methods for estimating similar parameters influence your analysis (inter-study variability)?

12. Some relations are not unexpected. The proportional relation between cell capacitance and cell size is obvious. Again, you should consider previous theoretical studies based on Rall's theory.

13. The authors should provide more clear evidence on how the proposed method should be translated to future studies on synaptic integration. Since the analysis did not consider any active property, I am not confident that the mathematical relations can help a more comprehensive computer simulation study.

14. Please use ACV or CV, not a mixture of both abbreviations.

15. The section on specific resistance (starting on Line 536) should be expanded to consider the fact that specific resistance may be different in different parts of the neuron (see Public review comments). This is important for modelers who want to explicitly represent dendrites in their motoneuron models (either as a separate compartment, an equivalent cable or completely reconstructed trees).

16. The Discussion section on relevance for modelers contains the following statement: "This supports the conclusions that the relative voltage threshold is constant within the MN pool and that Ohms law is followed in MNs". This is not strictly correct. Gustafsson and Pinter (1984b) showed that voltage threshold tended to be lower in low input conductance cells with long AHPs (see their Figure 4C and D). Also, the voltage threshold predicted by Ohms Law from the product of the measured rheobase and input resistance Vth = Ith*R tended to be lower than the measured voltage threshold, suggesting the activation of an inward conductance near threshold. This requires a variant of Ohm's Law in which the effective resistance is voltage-dependent: Vth=Ith*R(v).

17. Finally, although the vast majority of readers will know that correlation does not imply causation, it is worth stating explicitly in the Discussion section on limitations.

18. p. 33, l. 721: Actually, you should consider that in some experimental conditions, large (high-threshold) MNs attain lower firing rates than small (low-threshold) MNs. Maybe you have to explain in what conditions large MNs attain high firing rates.

---

## [Author Response]

Essential revisions:1. The format of the paper should follow the structure of eLife. The legend of Figure 1 should describe very explicitly the steps used to create the final size-related datasets, maybe giving an example. The circles and symbols in Figure 2 should be larger.

The authors modified the manuscript as suggested by the Reviewer.

To answer both this and comment #10, Figure 1 and its caption were changed to display a concrete example of how the two successive {R; S;_MN_} and {i_th_; s_MN_} final datasets were generated. This concrete example made it possible to go into the detail of the explicit steps used to create these datasets (L.308 to L.322). Figure 2 was adapted, and the circles and symbols were made larger.

2. The Introduction should be more inclusive and respectful with previous studies. It is unfair to state that some inferences are "speculative" since some authors provide linear regression curves (Eccles et al., 1958, for instance) while others consider the theoretical analysis by Rall. Also, when mentioning computational models, since the present study is focused on motor neurons, consider citing the extensive literature on models of motor neurons (see p. 3, l. 51 and 52).

We carefully considered this important comment, echoed by Reviewer 3. It was definitively not our intention to be disrespectful to previous work. We did not wish to mean that the correlations between MN properties were speculative, which is of course not true according to the multiple empirical and theoretical reports from previous studies of significant correlations between MN properties. Our message was rather that, although global conclusions on the relative variations of two MN properties (A is positively correlated with B) were made in previous review papers, these reviews relied on scattered experimental data from independent studies that used different mathematical models to describe the association between the same two MN properties (see L.42 to L.46 of the new Introduction). This weakened the possibility of previous papers to reach quantitative relationships between the MN properties. To avoid misinterpretations and the impression of not well acknowledging previous work, the Introduction has been entirely reshaped. The Introduction was also shortened and oriented to a broader audience, as advised by Reviewer 1.

In more details, the 1^st^ and 3^rd^ paragraphs of the first version of the Introduction were re-written (now L.20 to L.38). This new paragraph provides an extensive list of the strong associations between MN properties that were provided in previous review papers. Contrary to the original version, it is also now underlined that these empirical associations are consistent with the theoretical functions from Rall's cable theory, as extensively reviewed in Powers and Binder (2001), and detailed later in the Discussion section.

Then, a new section was added to the Introduction Section (L.39 to L.54). This section highlights the current limitations in the field that make it difficult to derive quantitative relationships between MN properties. Finally, the end of the introduction was condensed to the main outcomes and applications of this study (L.55 to L.74).

As detailed later in this document, the other sections of the manuscript that were not inclusive with previous studies, such as the 1^st^ section of the Discussion section, were also adjusted.

To follow Reviewer 1’s advice of shortening the Introduction, less emphasis was given on computational MN models in the Introduction; therefore, the authors consider that an extensive citation of the literature on models of MNs is less suitable in this new version of the Introduction.

3. In both Intro and Discussion, the authors should be careful when declaring the relevance of the current study. Several statements cannot be supported by analysis, for example, the comments that the method "can accelerate future research in the behaviour of individual MNs" and that mathematical equations can be scaled "to estimate the MN property values that cannot be measured in humans in vivo". How can you scale the equations without prior information on human motor neuron electrophysiology and morphology?

Thank you for pointing this out. In both the Introduction (L.58 to L.68) and the Discussion sections (L.783 to L.810), the relevance of the study was narrowed for conciseness and clarity to the most pertinent applications of the mathematical relationships. We also provided an example of application from our own work. Debatable or less evident applications such as the ones cited above were removed from the manuscript.

4. The methods section could be expanded to explain some of the measured electrophysiological properties in more detail. Most of the direct measurements (ACV, AHP duration, RN, Ith) are fairly straightforward, with the possible exception of time constant (see comments in Public Review). However, the derived measurements need to be explained in the Methods section. Both estimates of specific membrane resistivity (Rm) and whole cell capacitance (C) can be estimated using an equivalent cable model of the motoneuron along with an estimate of electrotonic length, and the formulas for these should be stated (see Ulfhake and Kellerth, Brain Res, 1984 equations 2 and 3 and Gustafsson and Pinter, JPhysiol, 1984a, equation 1). It should be mentioned in Methods that whole cell capacitance can be used as an estimate of cell surface area by assuming a value for specific capacitance of 1 microF/cm2 and that these area estimates are roughly in line with direct measurements although they tend to be a bit high (see Figure 6 of Gustafsson and Pinter 1984a).

The authors thank the Reviewing Editor and Reviewer 2 for underlining this imbalance between the measurements of S_MN_ and the electrophysiological properties in the description of the experimental approaches and their sources of error.

To extend this section, the selected experimental studies were reviewed to add to the manuscript a detailed description (L.148 to L.177) of the experimental approaches these studies took to measure ACV, AHP, R, I_th_, *τ* and c. The main sources of error in measuring these properties, that were reported in the selected studies and in other studies cited in the manuscript, were identified to be a variable membrane leak conductance around the electrode, voltage-induced membrane nonlinearities, and manual curve fitting, and are now detailed from L.178 to L.217. The equation taken from Gustafsson and Pinter (1984a) to C(C=τR⋅Ltanh⁡(L))

was added in L.177, in the context of an equivalent cable model.

As the values of R_m_ and the indirect predictions of s_neuron_ from electrophysiological values (Gustafsson and Pinter (1984a)) were not included in the global datasets, the two other equations the reviewer refers to

Rm=R⋅Sneuron⋅tanh⁡(L)L and Sneuron= CCm

were added in the Discussion section, where they were discussed as advised by the Reviewer. From L.712 to L.739, it is explained how Rm=R⋅Sneuron⋅tanh(L)L is consistent with the s_MN_ − ACV − AHP − R − I − C − *τ* relationships in Table 4 considering other findings reported in the literature, after which the relationships between these properties and r_m_ were calculated and added to Table 4, as done in the original version of the study. From L.757 to L.760, it is now discussed that the Sneuron\ =CCm relationship can be used to approximate the MN area if the value of c is known and an adequate constant value for c_m_ is taken.

To maintain consistency in the revised manuscript and clarify the Results section, the first sub-section of the Results section of the original version of the manuscript was moved to the Methods section from L.125 to L.147. This block, which has been unchanged since V1, covers the protocols taken by the selected studies to prepare the animals for experimentations and to perform morphometric measurements. It now introduces the added block, described above, about the electrophysiological measurements (from L.148 to L.177).

Sections of this answer were reproduced to answer Reviewer 2’s comments, with the intent of being made public with the preprint.

5. Data used in the analysis are both from in vivo and in vitro recordings, not exclusively from in vivo experiments as declared in several parts of the manuscript (e.g., Abstract). Morphometric data cannot be recorded in vivo.

The authors thank the Reviewing Editor for underlining this important source of confusion for the reader. In the first version of the manuscript, in vivo implicitly referred exclusively to the electrophysiological properties, and not to the morphological parameters which are indeed only obtainable in vitro. in vivo was being repeated through the manuscript to underline that the experimental studies reporting in vitro electrophysiological data were not included in this study. Where this could have led to some confusion, i.e. referred to in vivo morphological data, the related sentences were clarified, or the term ‘in vivo’ was removed in the revised version of the manuscript.

6. The authors should revise their method to include some information on data variability. A simple way is to provide the confidence interval of the regression. In this case, all parameters of the mathematical equations will have a range (95% confidence interval). The values provided in the current manuscript are only average values.

This is an important comment, in line with the comments from Reviewer 3 about data variability, as the inter-study data variability might have been a crucial factor in this analysis. Therefore, we considered important to provide a detailed analysis of data variability, that is now described in the Methods section from L.255 to L.279 and applied in the Results section from L.496 to L.533. The results of this analysis, which include the 95% confidence intervals the reviewer mentions, show in the revised manuscript that the data variability was low between the input experimental datasets and between the global datasets. These results are provided in Appendix Figures 1 to 4 of Appendix 1, by adding the ranges of the parameters for the 95% confidence interval in Table 3, and by plotting the 95% confidence intervals in Figure 3. Below we provide a summary of the procedures we followed.

First, we assessed the variability of the normalized property distributions between the experimental studies that constituted the same global dataset, for each of the 8 global datasets that included at least 2 experimental studies, i.e. {ACV; S_MN_}, {AHP; ACV}, {R; S_MN_}, {R; ACV}, {R; AHP}, {I_th_; R}, {I_th_; ACV}. Therefore, the inter-study data variability of 16 property distributions was analysed. ANOVA and Leven’s tests were not retained to perform this analysis of data variability as most of the property distributions were strongly not normal, according to a Jarque-Bera test for data normality which rejected the null hypothesis with p-values as low as 10^−9^. The Jarque-Bera test was preferred to other common tests (Shapiro-Wilk for instance) as it was identified to perform well for slightly skewed distributions with long tails (Thadewald, T., and Büning, H. (2007) Journal of applied statistics, 34(1), 87-105), which corresponds well to the data distributions in Figure FSM1. It was rather assessed for each global dataset whether the constitutive property distributions showed high inter-dataset variability in (1) the dataset size, (2) the range of the measured properties, (3) the mean of the distributions, (4) the dispersion of the distributions around their mean, and (5) the ratio between mean and median values of the distributions. The methodological approach to assess such variability was added to the Methods section. With the calculation of the standard deviations across the constitutive datasets of four metrics that described points (2) to (5) above, it could be concluded that in all global datasets, the constitutive studies (1) provided enough data points to have a significant impact on the regressions fitted to the global datasets, and reported data distribution that (2) spanned over similar normalized property ranges, (3) were centred around similar means, (4) displayed similar data dispersion, and (5) displayed similar skewness, as visually displayed with frequency histograms in Appendix Figure 1 of Appendix 1. These results were added to the Results section and the related graphs were added to the Appendix 1 section in Appendix Figure 2 and Appendix Figure 3. The 95% confidence intervals of the regressions were then added to the plots in Figure 2, and the ranges of the parameters of the mathematical equations added to Table 3. The reported narrow ranges of these intervals and the low variability of the regression parameters demonstrated that the constitutive studies showed little interstudy variability in associating the MN property distributions. The low observed inter-study variability made it pertinent to merge the normalized data from the selected studies into global datasets.

Further, we considered important to assess the variability between the distributions of the properties that appeared in different global datasets, for example the variability of the I_th_ distributions between the global datasets {I_th_; R}, {I_th_; ACV}, {I_th_; AHP} and {*τ*; I_th_}. This was assessed with the standard deviation across global datasets of the same four metrics indicated before, as now described in the Methods Section. Low variability was observed, which made the global datasets fit to be processed and transformed into final size-dependent datasets.

7. It is not clear why the authors used a 70-30% scheme for cross-validation and why only 5 validations were performed. A larger set of validations are probably needed and at least another training-testing proportion should be carried out.

Thank you for this comment. In the previous version of the manuscript, 5 iterations of 70-30 random splits of the global datasets were performed to constitute complementary training and test sets. The normalized relationships were derived from the training sets and validated against the test sets with three metrics (nME, NRMSE, r^2^) which were averaged across the five trials.

Rather than pursuing with this non-standard approach of randomly 70-30 splitting the data between the 5 iterations and improving this approach by repeating the shuffle-and-split process with different train-test ratios or an increased number of random splits, we have preferred to change the validation method and have opted for a standard and widely-accepted K-fold cross-validation approach, which is now described in the Methods from L.329 to L.349 and applied in the Results section from L.591 to L.599. The validation plots in Figure 6 were updated accordingly.

In brief, we chose K=5 for the K-fold cross-validation advised in ‘Deep Learning in Python’ from François Chollet (p. 87 section 3.6.4 Validating your approach using K-fold validation) when validating models on rather small datasets, as in our study (16 of the 17 datasets have less than 500 points). This 5-fold procedure is also the default validation method in the scikit-learn toolbox in Python. In this new approach, the data of each global dataset was shuffled once. Then, 5 non-overlapping partitions p_1_to p_5_ were created in each dataset, each containing 20% of the data (see Author response table 1). These partitions were permutated to create five different pairs of one training set including four partitions and one test set including one partition. For each permutation, and as before, the training set was used to derive the final relationships as described in the manuscript in the Methods. The final relationships were then validated against the corresponding test set. The final metrics (nME, nRMSE, r2) were averaged across the five permutations.

**Author response table 1. sa2table1:** 

For one {A; B} global dataset	Training set	Test set
Permutation 1	p_1_, p_2_, p_3_, p_4_	p_5_
Permutation 2	p_2_, p_3_, p_4_, p_5_	p_1_
Permutation 3	p_1_, p_3_, p_4_, p_5_	p_2_
Permutation 4	p_1_, p_2_, p_4_, p_5_	p_3_
Permutation 5	p_1_, p_2_, p_3_, p_5_	p_4_

The new validation results are strongly in agreement with the ones reported with the previous approach, as shown in Author response table 2.

**Author response table 2. sa2table2:** 

Datasets	Previous validation	New validation				
nME	nRMSE	R2	nME	nRMSE	R2
AHP = k_a_ ∙ S_MN_^a^	70	14	0,40	67	12	0,46
AHP = k_a_ ∙ ACV^a^	158	20	0,43	147	19	0,43
R = ka ∙ SMNa	255	20	0,56	312	31	0,51
R = k_a_ ∙ ACV^a^	246	23	0,43	205	18	0,46
R = k_a_ ∙ AHP^a^	153	21	0,63	143	21	0,66
Ith = ka ∙ Ra	410	19	0,37	300	19	0,36
I_th_ = k_a_ ∙ ACV^a^	171	26	0,34	155	20	0,42
I_th_ = k_a_ ∙ AHP^a^	84	15	0,59	78	17	0,61
C = k_a_ ∙ R^a^	61	12	0,58	55	13	0,57
C = ka ∙ ITaℎ	47	17	0,53	52	19	0,49
C = k_a_ ∙ AHP^a^	75	16	0,26	62	16	0,22
C = k_a_ ∙ ACV^a^	82	21	0,16	74	21	0,20
*τ* = k_a_ ∙ R^a^	86	13	0,54	73	13	0,51
*τ* = k_a_ ∙ AHP^a^	88	14	0,62	77	13	0,64
*τ* = ka ∙ ITaℎ	83	15	0,68	65	13	0,74
*τ* = k_a_ ∙ ACV^a^	81	17	0,38	84	16	0,44

8. Tables 4, 5 and 7 could be moved to the Supplementary material section.

We thank the Reviewing Editor and Reviewer 1 for their advice for a clearer and simpler framing of the paper. We modified the manuscript as suggested by the Reviewer and moved these three tables to Appendix 1, now Tables 3 to 5. The references in the text to these tables were adapted accordingly.

9. In the merged data sets shown in Figure 4 it would be useful to use different symbols for each data set (i.e., different symbols for {AHP;Smn}1 and {AHP;Smn}2).

Thank you for this comment which made Figure 4 more informative. The figure was adjusted, as suggested, and the sub-datasets constituting the final {A; S_MN_} datasets were identified with symbols specified in the caption of Figure 4 (L.576-577).

10. There is no oval box with fc(C) in Figure 1. Please, revise. Also, consider changing Figure 1 with a pseudo-code of the algorithm to improve clarity.

In line with the first comment of the reviewer, Figure 1 was revised to be a more detailed box-and-arrow diagram with a detailed caption explicating the step-by-step process applied to a specific example. The oval box issue was corrected.

11. Ith was estimated using different approaches in the selected studies. Some studies have used triangular-shaped currents, while others used step currents. How would different methods for estimating similar parameters influence your analysis (inter-study variability)?

As advised in the fourth comment from the Reviewing Editor and in the reviewer 2’s review report, we added in the revised manuscript a review of the experimental approaches taken by the selected studies for the measurements of the electrophysiological properties investigated in the manuscript (L.148 to L.177). When screening the selected studies, all the selected studies which provided I_th_ data for the global datasets in Figure 3 used step currents to estimate I_th_.

– Kernell (1966): ‘cells were stimulated to steady repetitive firing by long-lasting injected currents’

– Fleshman (1981): ‘Rheobase, defined as the minimal 50-ms current pulse necessary to produce an action potential, was measured by slowly increasing the current intensity until discharge occurred intermittently’.

– Kernell and Monster (1981): ‘The threshold current for maintained repetitive impulse firing was determined by aid of long-lasting injected currents’

– Gustafsson and Pinter (1984b): ‘The current magnitude necessary to depolarize from resting potential to the critical firing level using long (50-60 ms) current pulses injected through the recording micro-electrode’

– Zengel et al. (1985): ‘Rheobase was determined by slowly increasing the intensity of long (50 or 100 ms) current pulses until discharge occurred intermittently.’

– Munson et al. (1986) – Foehring et al. (1987): similar protocol as in Zengel et al. (1985)

– Krawitz et al. (2001): ‘Rheobase was defined as the minimum amplitude of a depolarizing (50 ms duration) current pulse that evoked an action potential.’

However, it must be noted that Krawitz et al., (2001) reported that ‘In two cells rheobase was estimated using a slowly rising current ramp’. However, the influence on the analysis of these two I_th_ data points being obtained with a different experimental approach, among the hundreds included in the global {I_th_; R} dataset, is negligeable. Therefore, this different experimental approach was not discussed in the manuscript.

12. Some relations are not unexpected. The proportional relation between cell capacitance and cell size is obvious. Again, you should consider previous theoretical studies based on Rall's theory.

Thank you for this comment, which echoes with Comment #4 and other comments by Reviewer 3. We realise that the Discussion section of the first version of the manuscript was not framed correctly and could have misled the reader to think that we were stating that we obtained unexpected and ‘new’ relationships when actually referring to well-known associations, such as the proportional relation between cell capacitance and cell size. We now avoid this issue in two ways. First, by providing in the Introduction section a list of well-known empirical associations, as discussed in Comment #2. And second by assessing in detail in the Discussion section in a rearranged and newly-named sub-section ‘Consistency of the relationships with previous empirical results and Rall’s theory’ (L.698 to L.760), the consistency of various combinations of the mathematical relationships in Table 4 with some of the equations that describe Rall’s theory of a MN equivalent cable model:

– C−Cm∙SMN

– τ=RmCm

– C=τR∙Ltanh(L)

– R=RmSneuron∙Ltanh(L)

Yet, we would like to highlight that the proportional relation between cell capacitance and cell size obtained in Table 4, while not unexpected in the literature, was not necessarily evident in the procedure taken in this study (L.748 to L.753). In the global datasets, the values of C were calculated as C=τR∙Ltanh(L) and empirically related to the four parameters ACV, AHP, R, I_th_, i.e. never to direct measures of S_MN_. Then, the inverted relationships S_MN_ = f(ACV, AHP, R, I_th_) obtained from 17 of the selected studies were used to derive the final {C; S_MN_} dataset, which happened to yield a proportional relationship, consistently with Rall’s theory, and without relying on direct measurements of S_MN_.

13. The authors should provide more clear evidence on how the proposed method should be translated to future studies on synaptic integration. Since the analysis did not consider any active property, I am not confident that the mathematical relations can help a more comprehensive computer simulation study.

This is an important comment concerning one of the proposed applications of this study. Consistently with one of Reviewer 3’s comments, we added in the Limitations section between L.924 and L.948 that this study mainly focused on passive MN electrophysiological properties; the membrane properties R, *τ*, C were measured with weak sub-threshold current pulses. Other properties like ACV, AHP, I_th_ were however measured in a functional context of elicited action potentials. Yet, some active MN membrane properties, such as PICs, were disregarded in this work, as they were not quantified in the selected studies simultaneously with the 9 MN properties investigated in this work. While quantitative relationships like those in Table 4 are missing for such active properties, the results from Table 4 can still be used in more comprehensive computer simulation studies involving Hodgkin-Huxley-like models for example, if they are linked with other observations in the literature that involve active properties, such as MNs of small S_MN_ having longer lasting total PICs of greater hyperpolarized activation than MNs of large S_MN_ (Heckman and Enoka, 2012).

The relationships in Table 4 can however be directly used in standard phenomenological MN models that typically neglect these active properties in a functional context, like the phenomenological RC LIF-like models Reviewer 3 refers to, which are known for their capacity to predict spiking behaviour (Teeter et al., 2018). Like in Watanabe et al., (2011) or Negro and Farina (2011) which involve comprehensive MN models, some previous studies using RC phenomenological models have attempted to build profiles of interconsistent MN-specific properties using maximum likelihood methods (Dong et al., 2011) and to describe a realistic continuous distribution of the MN properties in the MN pool (Negro et al., 2016) to estimate the proportion of common input to a motor neuron pool in humans. Such studies, and others that experimentally measured the distribution in the MN pool of some of the MN properties from Table 1 for modelling purposes (Raikova et al., 2018), would gain from the quantitative relationships provided in Table 4. An example of a direct application of Table 4 for modelling purposes is provided in Caillet et al., (2022) (https://www.biorxiv.org/content/10.1101/2022.02.21.481337v1.abstract), where the relationships in Table 4 are used to scale, after S_MN_ calibration, a cohort of LIF models to predict the firing behaviour of a theoretical MN pool from a subset of firing MNs identified with decomposed HDEMGs. The predicted spike trains then drive an N-MUs Hill-type model to predict the isometric muscle force trajectory.

14. Please use ACV or CV, not a mixture of both abbreviations.

Thank you for having spotted these inconsistencies. The manuscript was revised to maintain the ‘ACV’ abbreviation throughout the manuscript.

15. The section on specific resistance (starting on Line 536) should be expanded to consider the fact that specific resistance may be different in different parts of the neuron (see Public review comments). This is important for modelers who want to explicitly represent dendrites in their motoneuron models (either as a separate compartment, an equivalent cable or completely reconstructed trees).

Both this and Reviewer 2’s are important comments because a non-constant R_r_ value across the MN membrane is a limitation to using Rall’s simplest model of a MN (soma + dendrites) as a unique equivalent cylinder of constant diameter. In this view, the assumption of isopotentiality across the membrane surface was repeated in the manuscript whenever Rall’s approach was considered. We now discuss the variability of R_r_ across the somatodendritic surface and the subsequent limitations to the relationships in Table 4 in the Discussion section from L.731 to L.739 and in the Limitations Section from L.874 to L.880, with mention to MN modelling.

It should be noted that, after the comments from Reviewer 1 for a simpler framing of the paper, this section on R_r_ was moved from the Results section to the Discussion section (L.713 to L.739). In this updated paragraph, some sentences were left unchanged since the original version, others were rephrased or added.

16. The Discussion section on relevance for modelers contains the following statement: "This supports the conclusions that the relative voltage threshold is constant within the MN pool and that Ohms law is followed in MNs". This is not strictly correct. Gustafsson and Pinter (1984b) showed that voltage threshold tended to be lower in low input conductance cells with long AHPs (see their Figure 4C and D). Also, the voltage threshold predicted by Ohms Law from the product of the measured rheobase and input resistance Vth = Ith*R tended to be lower than the measured voltage threshold, suggesting the activation of an inward conductance near threshold. This requires a variant of Ohm's Law in which the effective resistance is voltage-dependent: Vth=Ith*R(v).

The authors thank the Reviewer for this comment that helped in enriching the discussion of the relationships in Table 4 (L.761 and L.781 of the Discussion section). It is now stated that, consistently with the relationships in Table 4 and in first approximation, Ohm’s law is followed in MNs when MNs are excited by weak subthreshold currents. This is how R was measured in the selected studies to avoid voltage-activated membrane nonlinearities, as discussed in the Methods section. It is then added in the manuscript that V_th_ may however be MN size-dependent in the MN pool, whatever the strength of the excitation current. It is finally mentioned that V_th_ is underestimated by the product RI_th_ near and above threshold, due to voltage-activated inward conductance. It is concluded that Ohm’s law must be revised both in the passive and the functional context to be a function of MN voltage and size.

17. Finally, although the vast majority of readers will know that correlation does not imply causation, it is worth stating explicitly in the Discussion section on limitations.

As advised by the Reviewer, this was added as the concluding limitation of the Limitations Section. L.949951:

‘As a final limitation, the relationships in Table 4 were obtained from a regression analysis and therefore provide correlations between some MN properties in a MN pool, but cannot be used to draw conclusions on the causality behind these associations.’

18. p. 33, l. 721: Actually, you should consider that in some experimental conditions, large (high-threshold) MNs attain lower firing rates than small (low-threshold) MNs. Maybe you have to explain in what conditions large MNs attain high firing rates.

Thank you for this comment. To remove any confusion regarding the ‘onion skin’ effect, it was added L.841843 that large MNs could attain higher firing rates than small MNs in the events of ballistic contractions or force generations close to maximum voluntary contractions.

References

Caillet, A., Phillips, A. T., Farina, D., and Modenese, L. (2022). Estimation of the firing behaviour of a complete motoneuron pool by combining EMG signal decomposition and realistic motoneuron modelling. bioRxiv.

Chollet F (2021) Deep Learning with Python. Simon and Schuster

Dong, Y., Mihalas, S., Russell, A., Etienne-Cummings, R., and Niebur, E. (2011). Estimating parameters of generalized integrate-and-fire neurons from the maximum likelihood of spike trains. Neural computation, 23(11), 2833-2867.

Eccles JC, Eccles RM, Lundberg A (1958) The action potentials of the α motoneurones supplying fast and slow muscles. J.Physiol.(Lond.) 142:275

Fleshman JW, Segev I, Burke RB (1988) Electrotonic architecture of type-identified α-motoneurons in the cat spinal cord. J.Neurophysiol. 60:60-85

Fleshman JW, Munson JB, Sypert GW, Friedman WA (1981) Rheobase, input resistance, and motor-unit type in medial gastrocnemius motoneurons in the cat. J Neurophysiol 46:1326-1338

Foehring RC, Sypert GW, Munson JB (1987) Motor-unit properties following cross-reinnervation of cat lateral gastrocnemius and soleus muscles with medial gastrocnemius nerve. II. influence of muscle on motoneurons. J Neurophysiol 57:1227-1245.

Gustafsson, Pinter MJ (1984a) An investigation of threshold properties among cat spinal α‐motoneurones. J. Physiol. 357:453-483.

Gustafsson, Pinter MJ (1984b) Relations among passive electrical properties of lumbar α-motoneurones of the cat. J. Physiol. 356:401-431.

Heckman CJ, Enoka RM (2012) Motor unit. Compr. Physiol. 2:2629-2682

Huh, S.,Heckman, C.J.,Manuel, M.(2021) Time course of alterations in adult spinal motoneuron properties in the SOD1(G93A) mouse model of ALS. eNeuro, 8, ENEURO

Kernell D, Monster AW (1981) Threshold current for repetitive impulse firing in motoneurones innervating muscle fibres of different fatigue sensitivity in the cat. Brain Res. 229:193-196

Kernell D (1966) Input resistance, electrical excitability, and size of ventral horn cells in cat spinal cord. Science 152:1637-1640

Krawitz S, Fedirchuk B, Dai Y, Jordan LM, McCrea DA (2001) State-dependent hyperpolarization of voltage threshold enhances motoneurone excitability during fictive locomotion in the cat. The Journal of physiology 532:271-281

Krutki P, Hałuszka A, Mrówczyński W, Gardiner PF, Celichowski J (2015) Adaptations of motoneuron properties to chronic compensatory muscle overload. Journal of neurophysiology 113:2769-2777.

Manuel M, Iglesias C, Donnet M, Leroy F, Heckman CJ, Zytnicki D (2009) Fast kinetics, high-frequency oscillations, and subprimary firing range in adult mouse spinal motoneurons. The Journal of neuroscience 29:11246-11256

Munson JB, Foehring RC, Lofton SA, Zengel JE, Sypert GW (1986) Plasticity of medial gastrocnemius motor units following cordotomy in the cat. Journal of neurophysiology 55:1454

Negro F, Yavuz U, Farina D (2016) The human motor neuron pools receive a dominant slow‐varying common synaptic input. J. Physiol. 594:5491-5505.

Negro, F., and Farina, D. (2011). Linear transmission of cortical oscillations to the neural drive to muscles is mediated by common projections to populations of motoneurons in humans. The Journal of physiology, 589(3), 629-637.

Powers RK, Binder MD (2001) Input-output functions of mammalian motoneurons. Rev. Physiol. Biochem. Pharmacol. 143:137-263

Raikova, R., Celichowski, J., Angelova, S., and Krutki, P. (2018). A model of the rat medial gastrocnemius muscle based on inputs to motoneurons and on an algorithm for prediction of the motor unit force. Journal of Neurophysiology, 120(4), 1973-1987.

Sasaki M (1991) Membrane properties of external urethral and external anal sphincter motoneurones in the cat. The Journal of physiology 440:345-366

Teeter C, Iyer R, Menon V, Gouwens N, Feng D, Berg J, Szafer A, Cain N, Zeng H, Hawrylycz M, Koch C, Mihalas S

(2018) Generalized leaky integrate-and-fire models classify multiple neuron types. Nat Commun 9:1-15

Ulfhake B, Kellerth JO (1984) Electrophysiological and morphological measurements in cat gastrocnemius and soleus α-motoneurones. Brain Res. 307:167-179

Watanabe, R. N., Magalhães, F. H., Elias, L. A., Chaud, V. M., Mello, E. M., and Kohn, A. F. (2013). Influences of premotoneuronal command statistics on the scaling of motor output variability during isometric plantar flexion. Journal of neurophysiology, 110(11), 2592-2606.

Zengel JE, Reid SA, Sypert GW, Munson JB (1985) Membrane electrical properties and prediction of motor-unit type of medial gastrocnemius motoneurons in the cat. J Neurophysiol 53:1323-1344